# Simple and Principled Uncertainty Estimation with Deterministic Deep Learning via Distance Awareness

**Jeremiah Zhe Liu**[*]
Google Research & Harvard University
jereliu@google.com

**Zi Lin**[†]
Google Research
lzi@google.com

**Shreyas Padhy**[†]
Google Research
shreyaspadhy@google.com

**Dustin Tran**
Google Research
trandustin@google.com

**Tania Bedrax-Weiss**
Google Research
tbedrax@google.com

**Balaji Lakshminarayanan**
Google Research
balajiln@google.com

## Abstract

Bayesian neural networks and deep ensembles are principled approaches to estimate the predictive uncertainty of a deep learning model. However their practicality in real-time, industrial-scale applications are limited due to their heavy memory and inference cost. This motivates us to study principled approaches to high-quality uncertainty estimation that require only a single deep neural network (DNN). By formalizing the uncertainty quantification as a minimax learning problem, we first identify *distance awareness*, i.e., the model's ability to properly quantify the distance of a testing example from the training data manifold, as a necessary condition for a DNN to achieve high-quality (i.e., minimax optimal) uncertainty estimation. We then propose *Spectral-normalized Neural Gaussian Process (SNGP)*, a simple method that improves the distance-awareness ability of modern DNNs, by adding a weight normalization step during training and replacing the output layer with a Gaussian Process. On a suite of vision and language understanding tasks and on modern architectures (Wide-ResNet and BERT), SNGP is competitive with deep ensembles in prediction, calibration and out-of-domain detection, and outperforms the other single-model approaches.[3]

## 1 Introduction

Efficient methods that reliably quantify a deep neural network (DNN)'s predictive uncertainty are important for industrial-scale, real-world applications, which include examples such as object recognition in autonomous driving [22], ad click prediction in online advertising [76], and intent understanding in a conversational system [84]. For example, for a natural language understanding (NLU) model built for a domain-specific chatbot service (e.g, weather inquiry), the user's input utterance to the model can be of any topic, and the model needs to understand reliably and in real-time whether to abstain or to trigger one of its known APIs.

When deep classifiers make predictions on input examples that are far from the support of the training set, their performance can be arbitrarily bad [4, 14]. This motivates the need for methods that are aware of the distance between an input test example and previously seen training examples, so they can return a uniform (i.e., maximum entropy) distribution over output labels if the input is too far from the training set (i.e., the input is out-of-domain) [30]. Gaussian processes (GPs) with suitable kernels enjoy such a property. However, to apply Gaussian processes to a high-dimensional machine

---

[*]Work done at Google Research.
[†]Work done as an Google AI Resident.
[3]Code available at https://github.com/google/uncertainty-baselines/tree/master/baselines.

learning problem, it is usually necessary to perform some form of feature extraction or dimensionality reduction using a DNN. Ideally, the hidden representation of a DNN should reflect a meaningful distance in the data manifold (e.g., the semantic textual similarity between two sentences), such that this "distance aware" property is preserved. However, as we will show in the experiments, this is often not guaranteed for common deep learning models (cf. Figure 1).

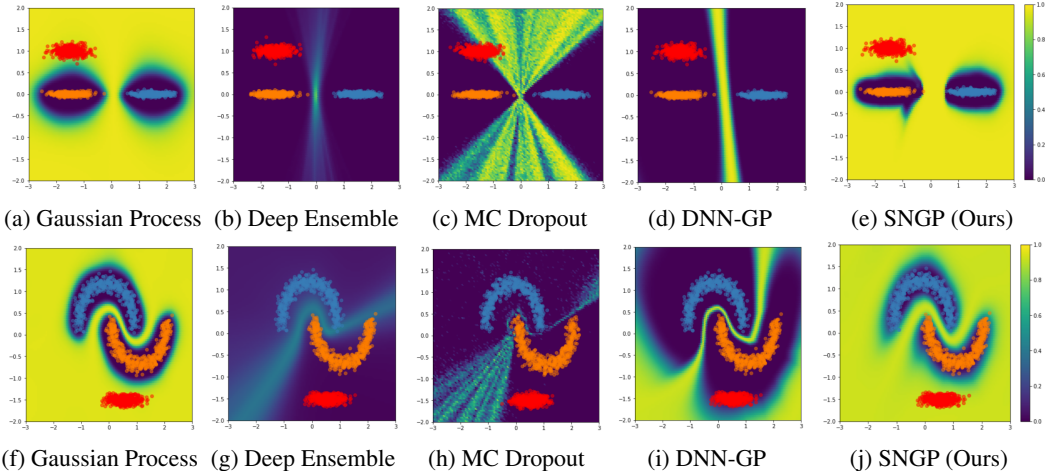

(a) Gaussian Process  (b) Deep Ensemble  (c) MC Dropout  (d) DNN-GP  (e) SNGP (Ours)

(f) Gaussian Process  (g) Deep Ensemble  (h) MC Dropout  (i) DNN-GP  (j) SNGP (Ours)

Figure 1: The uncertainty surface of a GP and different DNN approaches on the *two ovals* (Top Row) and *two moons* (Bottom Row) 2D classification benchmarks. SNGP is the only DNN-based approach achieving a distance-aware uncertainty similar to the gold-standard GP. Training data for positive (**Orange**) and negative classes (**Blue**). OOD data (**Red**) not observed during training. Background color represents the estimated model uncertainty (See 1e and 1j for color map). See Section 5.1 for details.

We propose a simple solution to this problem, namely adding *spectral normalization* to the weights in each (residual) layer [54]. We refer to our method as "Spectral-normalized Neural Gaussian Processes" (SNGP). We show that this provides bounds on $||h(\mathbf{x}) - h(\mathbf{x}')||_H$ relative to $||\mathbf{x} - \mathbf{x}'||_X$, where $\mathbf{x}$ and $\mathbf{x}'$ are two inputs, $h(\mathbf{x})$ is a deep feature extractor, and $||.||_X$ a semantically meaningful distance for the data manifold. We can then safely pass $h(\mathbf{x})$ into a distance-aware GP output layer. To ensure computational scalability, we approximate the GP posterior using a Laplace approximation to the random feature expansion of the GP, which gives rise to a model posterior that can be learned scalably and in closed-form with minimal modification to the training pipeline of a deterministic DNN, and allows us to efficiently compute the predictive uncertainty on a per-input basis without Monte Carlo sampling.

In the rest of this paper, we first theoretically motivate the importance of *distance awareness* for a model's ability uncertainty estimation by studying it as a minimax learning problem (Section 2). We then introduce our SNGP method in detail in Section 3, and experimentally evaluate its performance against other single-model approaches as well as deep ensembles in Section 5 [42]. On two challenging real world problems, namely image classification (using a Wide Resnet model on CIFAR-10 and CIFAR-100) and conversational intent understanding (using a BERT model on CLINC out-of-scope (OOS) intent dataset), we show that the SNGP method attains an uncertainty performance (e.g., calibration and out-of-domain (OOD) detection) that is competitive with that of a deep ensemble, while maintaining the accuracy and latency of a single deterministic DNN.

## 2 Distance Awareness: An Important Condition for High-Quality Uncertainty Estimation

**Notation and Problem Setup** Consider a data-generation distribution $p^*(y|\mathbf{x})$, where $y \in \{1, \dots, K\}$ is the space of $K$-class labels, and $\mathbf{x} \in \mathscr{X} \subset \mathbb{R}^d$ is the input data manifold equipped with a suitable metric $||.||_X$. In practice, the training data $\mathscr{D} = \{y_i, \mathbf{x}_i\}_{i=1}^N$ is often collected from a subset of the full input space $\mathscr{X}_{\text{IND}} \subset \mathscr{X}$. As a result, the full data-generating distribution $p^*(y|\mathbf{x})$ is in fact a mixture of an in-domain (IND) distribution $p_{\text{IND}}(y|\mathbf{x}) = p^*(y|\mathbf{x}, \mathbf{x} \in \mathscr{X}_{\text{IND}})$ and also an OOD distribution $p_{\text{OOD}}(y|\mathbf{x}) = p^*(y|\mathbf{x}, \mathbf{x} \notin \mathscr{X}_{\text{IND}})$ [52, 66]:

$$p^*(y|\mathbf{x}) = \qquad p^*(y, \mathbf{x} \in \mathscr{X}_{\text{IND}}|\mathbf{x}) \qquad + \qquad p^*(y, \mathbf{x} \notin \mathscr{X}_{\text{IND}}|\mathbf{x})$$
$$= p^*(y|\mathbf{x}, \mathbf{x} \in \mathscr{X}_{\text{IND}}) * p^*(\mathbf{x} \in \mathscr{X}_{\text{IND}}) + p^*(y|\mathbf{x}, \mathbf{x} \notin \mathscr{X}_{\text{IND}}) * p^*(\mathbf{x} \notin \mathscr{X}_{\text{IND}}). \tag{1}$$

During training, the model learns the in-domain distribution $p^*(y|\mathbf{x}, \mathbf{x} \in \mathscr{X}_{\mathtt{IND}})$ from the data $\mathscr{D}$, but does not have knowledge about $p^*(y|\mathbf{x}, \mathbf{x} \notin \mathscr{X}_{\mathtt{IND}})$. In the weather-service chatbot example, the out-of-domain space $\mathscr{X}_{\mathtt{OOD}} = \mathscr{X} / \mathscr{X}_{\mathtt{IND}}$ is the space of all natural utterances not related to weather queries, whose elements usually do not have a meaningful correspondence with the in-domain intent labels $y_k \in \{1, \ldots, K\}$. Therefore, the out-of-domain distribution $p^*(y|\mathbf{x}, \mathbf{x} \notin \mathscr{X}_{\mathtt{IND}})$ can be very different from the in-domain distribution $p^*(y|\mathbf{x}, \mathbf{x} \in \mathscr{X}_{\mathtt{IND}})$, and we only expect the model to generalize well within $\mathscr{X}_{\mathtt{IND}}$. However, during testing, the model needs to construct a predictive distribution $p(y|\mathbf{x})$ for the entire input space $\mathscr{X} = \mathscr{X}_{\mathtt{IND}} \cup \mathscr{X}_{\mathtt{OOD}}$, since the users' utterances can be of any topic.

## 2.1 Uncertainty Estimation as a Minimax Learning Problem

To formulate the uncertainty estimation as a learning problem under (1), we need to define a loss function to measure a model $p(y|\mathbf{x})$'s quality of predictive uncertainty. A popular uncertainty metric is Expected Calibration Error (ECE), defined as $C(p, p^*) = E\big[|E(y^* = \hat{y}|\hat{p} = p) - p|\big]$, which measures the difference in expectation between the model's predictive confidence (e.g., the maximum probability score) and its actual accuracy [29, 56]. However, ECE is not suitable as a loss function, since it is not uniquely minimized at $p = p^*$. Specifically, there can exist a trivial predictor that ignores the input example and achieves perfect calibration by predicting randomly according to the marginal distribution of the labels [24].

To this end, a theoretically more well-founded uncertainty metric is to examine *strictly proper scoring rules* [25] $s(., p^*)$, which is uniquely minimized by the true distribution $p = p^*$. Examples include log-loss and Brier score. Proper scoring rules are related to ECE in that it is an upper bound of the calibration error by the classic calibration-refinement decomposition [10]. Therefore, minimizing a proper scoring rule implies minimizing the calibration error of the model. Consequently, we can formalize the problem of uncertainty quantification as the problem of constructing an optimal predictive distribution $p(y|\mathbf{x})$ to minimize the expected risk over the entire $\mathbf{x} \in \mathscr{X}$, i.e., an *Uncertainty Risk Minimization* problem:

$$\inf_{p \in \mathscr{P}} S(p, p^*) = \inf_{p \in \mathscr{P}} E_{\mathbf{x} \in \mathscr{X}}\big[s(p, p^*|\mathbf{x})\big]. \tag{2}$$

Unfortunately, directly minimizing (2) over the entire input space $\mathscr{X}$ is not possible even with infinite amounts of data. This is because since the data is collected only from $\mathscr{X}_{\mathtt{IND}}$, the true OOD distribution $p^*(y|\mathbf{x}, \mathbf{x} \notin \mathscr{X}_{\mathtt{IND}})$ is never learned by the model, and generalization is not guaranteed since $p^*(y|\mathbf{x}, \mathbf{x} \in \mathscr{X}_{\mathtt{IND}})$ and $p^*(y|\mathbf{x}, \mathbf{x} \notin \mathscr{X}_{\mathtt{IND}})$ are not assumed to be similar. As a result, the naive practice of using a model trained only with in-domain data to generate OOD predictions can lead to arbitrarily bad results, since nature can happen to produce an OOD distribution $p^*(y|\mathbf{x}, \mathbf{x} \notin \mathscr{X}_{\mathtt{IND}})$ that is at odds with the model prediction. This is clearly undesirable for safety-critical applications. To this end, a more prudent strategy is to instead minimize the *worst-case* risk with respect to all possible $p^* \in \mathscr{P}^*$, i.e., construct $p(y|x)$ to minimize the *Minimax Uncertainty Risk*:

$$\inf_{p \in \mathscr{P}} \Big[\sup_{p^* \in \mathscr{P}^*} S(p, p^*)\Big]. \tag{3}$$

In game-theoretic nomenclature, the uncertainty estimation problem acts as a two-player game of model v.s. nature, where the goal of the model is to produce a minimax strategy $p$ that minimizes the risk $S(p, p^*)$ against all possible (even adversarial) moves $p^*$ of nature. Under the classification task and for Brier score, the solution to the minimax problem (3) adopts a simple and elegant form:

$$p(y|\mathbf{x}) = p(y|\mathbf{x}, \mathbf{x} \in \mathscr{X}_{\mathtt{IND}}) * p^*(\mathbf{x} \in \mathscr{X}_{\mathtt{IND}}) + p_{\mathtt{uniform}}(y|\mathbf{x}, \mathbf{x} \notin \mathscr{X}_{\mathtt{IND}}) * p^*(\mathbf{x} \notin \mathscr{X}_{\mathtt{IND}}). \tag{4}$$

This is very intuitive: if an input point is in the training data domain, trust the model, otherwise use a uniform (maximum entropy) prediction. For the practice of uncertainty estimation, (4) is conceptually important in that it verifies that there exists a unique optimal solution to the uncertainty estimation problem (3). Furthermore, this optimal solution can be constructed conveniently. Specifically, it can be constructed as a mixture of a discrete uniform distribution $p_{\mathtt{uniform}}$ and the in-domain predictive distribution $p(y|\mathbf{x}, \mathbf{x} \in \mathscr{X}_{\mathtt{IND}})$ that the model has already learned from data, *assuming one can quantify* $p^*(\mathbf{x} \in \mathscr{X}_{\mathtt{IND}})$ *well*. In fact, the expression (4) can be shown to be optimal for a broad family of scoring rules known as the Bregman score, which includes the Brier score and the widely used *log score* as the special cases. We derive (4) in Appendix B.

## 2.2 Input Distance Awareness as a Necessary Condition

In light of Equation (4), a key capacity for a deep learning model to reliably estimate predictive uncertainty is its ability to quantify, either explicitly or implicitly, the domain probability $p(\mathbf{x} \in \mathscr{X}_{\mathtt{IND}})$.

This requires the model to have a good notion of the distance (or dissimilarity) between a testing example $\mathbf{x}$ and the training data $\mathscr{X}_{\text{IND}}$ with respect to a *meaningful* distance $||.||_X$ for the data manifold (e.g., semantic textual similarity [12] for language data). Definition 1 makes this notion more precise:

**Definition 1** (Input Distance Awareness). *Consider a predictive distribution $p(y|\mathbf{x})$ trained on a domain $\mathscr{X}_{\text{IND}} \subset \mathscr{X}$, where $(\mathscr{X}, ||.||_X)$ is the input data manifold equipped with a suitable metric $||.||_X$. We say $p(y|\mathbf{x})$ is input distance aware if there exists $u(\mathbf{x})$ a summary statistic of $p(y|\mathbf{x})$ that quantifies model uncertainty (e.g., entropy, predictive variance, etc) that reflects the distance between $\mathbf{x}$ and the training data with respect to $||.||_X$, i.e.,*

$$u(\mathbf{x}) = v\big(d(\mathbf{x}, \mathscr{X}_{\text{IND}})\big)$$

*where $v$ is a monotonic function and $d(\mathbf{x}, \mathscr{X}_{\text{IND}}) = E_{\mathbf{x}' \sim \mathscr{X}_{\text{IND}}}||\mathbf{x} - \mathbf{x}'||_X^2$. is the distance between $\mathbf{x}$ and the training data domain.*

A classic model that satisfies the *distance-awareness* property is a Gaussian process (GP) with a radial basis function (RBF) kernel. Its predictive distribution $p(y|\mathbf{x}) = softmax(g(\mathbf{x}))$ is a softmax transformation of the GP posterior $g \sim GP$ under the cross-entropy likelihood, and its predictive uncertainty can be expressed by the posterior variance $u(\mathbf{x}^*) = var(g(\mathbf{x}^*)) = 1 - \mathbf{k}^{*\top}\mathbf{V}\mathbf{k}^*$ for $\mathbf{k}_i^* = exp(-\frac{1}{2l}||\mathbf{x}^* - \mathbf{x}_i||_X^2)$ and $\mathbf{V}_{N \times N}$ a fixed matrix determined by data. Then $u(\mathbf{x}^*)$ increases monotonically toward 1 as $\mathbf{x}^*$ moves further away from $\mathscr{X}_{\text{IND}}$ [61]. In view of the expression (4), the *input distance awareness* property is important for both calibration and OOD detection. However, this property is not guaranteed for a typical deep learning model [33]. Consider a discriminative deep classifier with dense output layer $logit_k(\mathbf{x}) = h(\mathbf{x})^\top \beta_k$, whose model confidence (i.e., maximum predictive probability) is characterized by the magnitude of the class logits, which is defined by the inner product distances between the hidden representation $h(\mathbf{x})$ and the decision boundaries $\{\beta_k\}_{k=1}^K$ (see, e.g., Figure 1b-1c and 1g-1h). As a result, the model computes confidence for a $\mathbf{x}^*$ based not on its distance from the training data $\mathscr{X}_{\text{IND}}$, but based on its distance from the decision boundaries, i.e., the model uncertainty is not *input distance aware*.

**Two Conditions for Input Distance Awareness in Deep Learning** Notice that a deep learning model $logit(\mathbf{x}) = g \circ h(\mathbf{x})$ is commonly composed of a hidden mapping $h : \mathscr{X} \to \mathscr{H}$ that maps the input $\mathbf{x}$ into a hidden representation space $h(\mathbf{x}) \in \mathscr{H}$, and an output layer $g$ that maps $h(\mathbf{x})$ to the label space. To this end, a DNN $logit(\mathbf{x}) = g \circ h(\mathbf{x})$ can be made *input distance aware* via a combination of two conditions: **(1)** make the output layer $g$ *distance aware*, so it outputs an uncertainty metric reflecting distance in the hidden space $||h(\mathbf{x}) - h(\mathbf{x}')||_H$ (in practice, this can be achieved by using a GP with a shift-invariant kernel as the output layer), and **(2)** make the hidden mapping *distance preserving* (defined below), so that the distance in the hidden space $||h(\mathbf{x}) - h(\mathbf{x}')||_H$ has a meaningful correspondence to the distance $||\mathbf{x} - \mathbf{x}'||_X$ in the data manifold. From the mathematical point of view, this is equivalent to requiring $h$ to satisfy the *bi-Lipschitz* condition [67]:

$$L_1 * ||\mathbf{x}_1 - \mathbf{x}_2||_X \le ||h(\mathbf{x}_1) - h(\mathbf{x}_2)||_H \le L_2 * ||\mathbf{x}_1 - \mathbf{x}_2||_X, \tag{5}$$

for positive and bounded constants $0 < L_1 < 1 < L_2$. It is worth noticing that for a deep learning model, the bi-Lipschitz condition (5) usually leads the model's hidden space to preserve a semantically meaningful distance in the input data manifold $\mathscr{X}$, rather than a naive metric such as the square distance in the pixel space. This is because that the upper Lipschitz bound $||h(\mathbf{x}_1) - h(\mathbf{x}_2)||_H \le L_2 * ||\mathbf{x}_1 - \mathbf{x}_2||_X$ is an important condition for the adversarial robustness of a deep network, which prevents the hidden representations $h(\mathbf{x})$ from being overly sensitive to the semantically meaningless perturbations in the pixel space [65, 80, 75, 37, 71]. On the other hand, the lower Lipschitz bound $||h(\mathbf{x}_1) - h(\mathbf{x}_2)||_H \ge L_1 * ||\mathbf{x}_1 - \mathbf{x}_2||_X$ prevents the hidden representation from being unnecessarily invariant to the semantically meaningful changes in the input manifold [38, 77]. Combined together, the bi-Lipschitz condition essentially encourages $h$ to be an approximately isometric mapping, thereby ensuring that the learned representation $h(\mathbf{x})$ has a robust and meaningful correspondence with the semantic properties of the input data $\mathbf{x}$. Although not stated explicitly, learning an approximately isometric and geometry-preserving mapping is a common goal in machine learning. For example, image classifiers strive to learn a mapping from image manifold to a hidden space that can be well-separated by a set of linear decision boundaries, and sentences encoders aim to project sentences into a vector space where the cosine distance reflects the semantic similarity in natural language. Finally, it is worth noting that preserving such approximate isometry in a neural network is possible even after significant dimensionality reduction [8, 32, 59, 64].

# 3 SNGP: A Simple Approach to Distance-aware Deep Learning

In this section we propose *Spectral-normalized Neural Gaussian Process (SNGP)*, a simple method to improve the *input distance awareness* ability of a modern residual-based DNN (e.g., ResNet, Transformer) by (1) making the output layer *distance aware* and (2) making the hidden layers *distance preserving*, as discussed in Section 2.2. Full method is summarized in Algorithms 1-2.

## 3.1 Distance-aware Output Layer via Laplace-approximated Neural Gaussian Process

To make the output layer $g : \mathcal{H} \to \mathcal{Y}$ distance aware, SNGP replaces the typical dense output layer with a Gaussian process (GP) with an RBF kernel, whose posterior variance at $\mathbf{x}^*$ is characterized by its $L_2$ distance from the training data in the hidden space. Specifically, given $N$ training samples $\mathcal{D} = \{y_i, \mathbf{x}_i\}_{i=1}^N$ and denoting $h_i = h(\mathbf{x}_i)$, the Gaussian-process output layer $g_{N\times 1} = [g(h_1), \dots, g(h_N)]^\top$ follows a multivariate normal distribution *a priori*:

$$g_{N\times 1} \sim MVN(\mathbf{0}_{N\times 1}, \mathbf{K}_{N\times N}), \text{where } \mathbf{K}_{i,j} = \exp(-||h_i - h_j||_2^2/2), \tag{6}$$

and the posterior distribution is computed as $p(g|\mathcal{D}) \propto p(\mathcal{D}|g)p(g)$ where $p(g)$ is the GP prior in (6) and $p(\mathcal{D}|g)$ is the data likelihood for classification (i.e., the exponentiated cross-entropy loss). However, computing the exact Gaussian process posterior for a large-scale classification task is both analytically intractable and computationally expensive, In this work, we propose a simple approximation strategy for GP that is based on a Laplace approximation to the random Fourier feature (RFF) expansion of the GP posterior [61]. Our approach gives rise to a closed-form posterior that is end-to-end trainable with the rest of the neural network, and empirically leads to an improved quality in estimating the posterior uncertainty. Specifically, we first approximate the GP prior in (6) by deploying a low-rank approximation to the kernel matrix $\mathbf{K} = \Phi\Phi^\top$ using random features [60]:

$$g_{N\times 1} \sim MVN(\mathbf{0}_{N\times 1}, \Phi\Phi_{N\times N}^\top), \qquad \text{where} \qquad \Phi_{i,D_L\times 1} = \sqrt{2/D_L} * \cos(-\mathbf{W}_L h_i + \mathbf{b}_L), \tag{7}$$

where $h_i = h(\mathbf{x}_i)$ is the hidden representation in the penultimate layer with dimension $D_{L-1}$. $\Phi_i$ is the final layer with dimension $D_L$, it contains $\mathbf{W}_{L,D_L\times D_{L-1}}$ a fixed weight matrix whose entries are sampled i.i.d. from $N(0,1)$, and $\mathbf{b}_{L,D_L\times 1}$ a fixed bias term whose entries are sampled i.i.d. from $Uniform(0, 2\pi)$. As a result, for the $k^{th}$ logit, the RFF approximation to the GP prior in (6) can be written as a neural network layer with fixed hidden weights $\mathbf{W}$ and learnable output weights $\beta_k$:

$$g_k(h_i) = \sqrt{2/D_L} * \cos(-\mathbf{W}_L h_i + \mathbf{b}_L)^\top \beta_k, \qquad \text{with prior} \qquad \beta_{k,D_L\times 1} \sim N(0, \mathbf{I}_{D_L\times D_L}). \tag{8}$$

Notice that conditional on $h$, $\beta = \{\beta_k\}_{k=1}^K$ is the only learnable parameter in the model. As a result, the RFF approximation in (8) reduces an infinite-dimensional GP to a standard Bayesian linear model, for which many posterior approximation methods (e.g., expectation propagation (EP)) can be applied [53]. In this work, we choose the Laplace method due to its simplicity and the fact that its posterior variance has a convenient closed form [61]. Briefly, the Laplace method approximates the RFF posterior $p(\beta|\mathcal{D})$ using a Gaussian likelihood centered around the maximum a posterior (MAP) estimate $\hat{\beta} = \text{argmax}_\beta \, p(\beta|\mathcal{D})$, such that $p(\beta_k|\mathcal{D}) \approx MVN(\hat{\beta}_k, \hat{\Sigma}_k = \hat{\mathbf{H}}_k^{-1})$, where $\hat{\mathbf{H}}_{k,(i,j)} = \frac{\partial^2}{\partial \beta_i \partial \beta_j} \log p(\beta_k|\mathcal{D})|_{\beta_k=\hat{\beta}_k}$ is the $D_L \times D_L$ Hessian matrix of the log posterior likelihood evaluated at the MAP estimates. Under the linear-model formulation of the RFF posterior, the posterior precision matrix (i.e., the inverse covariance matrix) adopts a simple expression $\hat{\Sigma}_k^{-1} = \mathbf{I} + \sum_{i=1}^N \hat{p}_{i,k}(1 - \hat{p}_{i,k})\Phi_i\Phi_i^\top$, where $p_{i,k}$ is the model prediction $softmax(\hat{g}_i)$ under the MAP estimates $\hat{\beta} = \{\beta_k\}_{k=1}^K$ [61]. To summarize, the Laplace posterior for GP under the RFF approximation is:

$$\beta_k|\mathcal{D} \sim MVN(\hat{\beta}_k, \hat{\Sigma}_k), \quad \text{where} \quad \hat{\Sigma}_k^{-1} = \mathbf{I} + \sum_{i=1}^N \hat{p}_{i,k}(1 - \hat{p}_{i,k})\Phi_i\Phi_i^\top. \tag{9}$$

During minibatch training, the posterior mean $\hat{\beta}$ is updated via regular stochastic gradient descent (SGD) with respect to the (unnormalized) log posterior $-\log p(\beta|\mathcal{D}) = -\log p(\mathcal{D}|\beta) + \frac{1}{2}||\beta||^2$ where $-\log p(\mathcal{D}|\beta)$ is the cross-entropy loss. The posterior precision matrix is updated cheaply as $\hat{\Sigma}_{k,t}^{-1} = (1-m) * \hat{\Sigma}_{k,t-1}^{-1} + m * \sum_{i=1}^M \hat{p}_{i,k}(1 - \hat{p}_{i,k})\Phi_i\Phi_i^\top$ for a minibatch of size $M$ and $m$ a small scaling coefficient. This computation only needs to be performed by passing through training data once at the final epoch. As a result, the GP posterior (9) can be learned scalably and in closed-form with minimal modification to the training pipeline of a deterministic DNN. It is worth noting that the Laplace approximation to the RFF posterior is asymptotically exact by the virtue of the Bernstein-von Mises (BvM) theorem and the fact that (8) is a finite-rank model [16, 23, 46, 57].

## 3.2 Distance-preserving Hidden Mapping via Spectral Normalization

Replacing the output layer $g$ with a Gaussian process only allows the model $logit(\mathbf{x}) = g \circ h(\mathbf{x})$ to be aware of the distance in the hidden space $||h(\mathbf{x}_1) - h(\mathbf{x}_2)||_H$. It is also important to ensure the hidden mapping $h$ is *distance preserving* so that the distance in the hidden space $||h(\mathbf{x}) - h(\mathbf{x}')||_H$ has a meaningful correspondence to the distance in the input space $||\mathbf{x} - \mathbf{x}'||_X$. To this end, we notice that modern deep learning models (e.g., ResNets, Transformers) are commonly composed of residual blocks, i.e., $h(\mathbf{x}) = h_{L-1} \circ \cdots \circ h_2 \circ h_1(\mathbf{x})$ where $h_l(\mathbf{x}) = \mathbf{x} + g_l(\mathbf{x})$. For such models, there exists a simple method to ensure $h$ is *distance preserving*: by bounding the Lipschitz constants of all nonlinear residual mappings $\{g_l\}_{l=1}^{L-1}$ to be less than 1. We state this result formally below:

**Proposition 1** (Lipschitz-bounded residual block is distance preserving [3]). *Consider a hidden mapping $h: \mathcal{X} \to \mathcal{H}$ with residual architecture $h = h_{L-1} \circ \ldots h_2 \circ h_1$ where $h_l(\mathbf{x}) = \mathbf{x} + g_l(\mathbf{x})$. If for $0 < \alpha \leq 1$, all $g_l$'s are $\alpha$-Lipschitz, i.e., $||g_l(\mathbf{x}) - g_l(\mathbf{x}')||_H \leq \alpha ||\mathbf{x} - \mathbf{x}'||_X \quad \forall (\mathbf{x}, \mathbf{x}') \in \mathcal{X}$. Then:*

$$L_1 * ||\mathbf{x} - \mathbf{x}'||_X \leq ||h(\mathbf{x}) - h(\mathbf{x}')||_H \leq L_2 * ||\mathbf{x} - \mathbf{x}'||_X,$$

*where $L_1 = (1 - \alpha)^{L-1}$ and $L_2 = (1 + \alpha)^{L-1}$, i.e., h is distance preserving.*

Proof is in Appendix E.1. The ability of a residual network to construct a geometry-preserving metric transform between the input space $\mathcal{X}$ and the hidden space $\mathcal{H}$ is well-established in learning theory and generative modeling literature, but the application of these results in the context of uncertainty estimation for DNN appears to be new [3, 5, 32, 64].

Consequently, to ensure the hidden mapping $h$ is distance preserving, it is sufficient to ensure that the weight matrices for the nonlinear residual block $g_l(\mathbf{x}) = \sigma(\mathbf{W}_l\mathbf{x} + \mathbf{b}_l)$ to have spectral norm (i.e., the largest singular value) less than 1, since $||g_l||_{Lip} \leq ||\mathbf{W}_l\mathbf{x} + \mathbf{b}_l||_{Lip} \leq ||\mathbf{W}_l||_2 \leq 1$. In this work, we enforce the aforementioned Lipschitz constraint on $g_l$'s by applying the *spectral normalization (SN)* on the weight matrices $\{\mathbf{W}_l\}_{l=1}^{L-1}$ as recommended in [5]. Briefly, at every training step, the SN method first estimate the spectral norm $\hat{\lambda} \approx ||\mathbf{W}_l||_2$ using the power iteration method [26, 54], and then normalizes the weights as:

$$\mathbf{W}_l = \begin{cases} c * \mathbf{W}_l/\hat{\lambda} & \text{if } c < \hat{\lambda} \\ \mathbf{W}_l & \text{otherwise} \end{cases} \tag{10}$$

where $c > 0$ is a hyperparameter used to adjust the exact spectral norm upper bound on $||\mathbf{W}_l||_2$ (so that $||\mathbf{W}_l||_2 \leq c$). This hyperparameter is useful in practice since the other regularization mechanisms (e.g., Dropout, Batch Normalization) in the hidden layers can rescale the Lipschitz constant of the original residual mapping [26]. Therefore, (10) allows us more flexibility in controlling the spectral norm of the neural network weights so it is the most compatible with the architecture at hand.

**Method Summary** We summarize the method in Algorithms 1-2. As shown, for every minibatch step, the model first updates the hidden-layer weights $\{\mathbf{W}_l, \mathbf{b}_l\}_{l=1}^{L-1}$ and the trainable output weights $\beta = \{\beta_k\}_{k=1}^K$ via SGD, then performs spectral normalization, and finally (if in the final epoch) performs precision matrix update (Equation (9)). We discuss further details (e.g. computational complexity) in Appendix A.

---

**Algorithm 1** SNGP Training

1: **Input:**
  Minibatches $\{D_i\}_{i=1}^N$ for $D_i = \{y_m, \mathbf{x}_m\}_{m=1}^M$.
2: **Initialize:**

  $\hat{\Sigma} = \mathbf{I}, \mathbf{W}_L \overset{iid}{\sim} N(0,1), \mathbf{b}_L \overset{iid}{\sim} U(0, 2\pi)$

3: **for** train_step = 1 **to** max_step **do**
4:   SGD update $\left\{\beta, \{\mathbf{W}_l\}_{l=1}^{L-1}, \{\mathbf{b}_l\}_{l=1}^{L-1}\right\}$
5:   Spectral Normalization $\{\mathbf{W}_l\}_{l=1}^{L-1}$ (10).
6:   **if** final_epoch **then**
7:     Update precision matrix $\{\hat{\Sigma}_k^{-1}\}_{k=1}^K$ (9).
8:   **end if**
9: **end for**
10: Compute posterior covariance $\hat{\Sigma}_k = inv(\hat{\Sigma}_k^{-1})$.

---

**Algorithm 2** SNGP Prediction

1: **Input:** Testing example $\mathbf{x}$.
2: Compute Feature:

  $\Phi_{D_L \times 1} = \sqrt{2/D_L} * \cos(\mathbf{W}_L h(\mathbf{x}) + \mathbf{b}_L),$

3: Compute Posterior Mean:

  $\text{logit}_k(\mathbf{x}) = \Phi^\top \beta_k$

4: Compute Posterior Variance:

  $\text{var}_k(\mathbf{x}) = \Phi^\top \hat{\Sigma}_k \Phi.$

5: Compute Predictive Distribution:

  $p(y|\mathbf{x}) = \int_{m \sim N(\text{logit}(\mathbf{x}), \text{var}(\mathbf{x}))} \text{softmax}(m)$

---

## 4 Related Work

**Single-model approaches to deep classifier uncertainty** Recent work examines uncertainty methods that add few additional parameters or runtime cost to the base model. The state-of-the-art on large-scale tasks are efficient ensemble methods [79, 21], which cast a set of models under a single one, encouraging independent member predictions using low-rank perturbations. These methods are parameter-efficient but still require multiple forward passes from the model. SNGP investigates an orthogonal approach that improves the uncertainty quantification by imposing suitable regularization on a single model, and therefore requires only a single forward pass during inference. There exists other runtime-efficient, single-model approaches to estimate predictive uncertainty, achieved by either replacing the loss function [33, 50, 51, 68, 69], the output layer [6, 72, 11, 48], or computing a closed-form posterior for the output layer [62, 70, 41]. SNGP builds on these approaches by also considering the intermediate representations which are necessary for good uncertainty estimation, and proposes a simple method (spectral normalization) to achieve it. A recent method named Deterministic Uncertainty Quantification (**DUQ**) also regulates the neural network mapping but uses a two-sided gradient penalty [77]. The two-sided gradient penalty can be undesirable for a residual network, since imposing $||\nabla f|| = 1$ onto a residual connection $f(\mathbf{x}) = \mathbf{x} + g(\mathbf{x})$ can force $g(\mathbf{x})$ toward 0, leading to an identity mapping. We compare with DUQ in our experiments.

**Laplace approximation and GP inference with DNN** Laplace approximation has a long history in GP and NN literature [73, 17, 61, 49, 63], and the theoretical connection between a Laplace-approximated DNN and GP has being explored recently [40]. Differing from these works, SNGP applies the Laplace approximation to the posterior of a neural GP, rather than to a shallow GP or a dense-output-layer DNN. Earlier works that combine a GP with a DNN usually perform MAP estimation [11] or structured Variational Inference (VI) [9, 81]. These approaches were shown to lead to poor calibration by recent work [74], which proposed a simple fix by combing Monte Carlo Dropout (MC Dropout) with random Fourier features, which we term Calibrated Deep Gaussian Process (**MCD-GP**). SNGP differs from MCD-GP in that it considered a different regularization approach (spectral normalization) and can compute its posterior uncertainty more efficiently in a single forward pass. We compare with MCD-GP in our experiments. Appendix D contains further related work on distance-preserving neural networks and open-set classification.

## 5 Experiments

### 5.1 2D Synthetic Benchmark

We first study the behavior of the uncertainty surface of a SNGP model under a suite of 2D classification benchmarks. Specifically, we consider the *two ovals* benchmark (Figure 1, row 1) and the *two moons* benchmark (Figure 1, row 2). The *two ovals* benchmark consists of two near-flat Gaussian distributions, which represent the two in-domain classes (orange and blue) that are separable by a linear decision boundary. There also exists an OOD distribution (red) that the model doesn't observe during training. Similarly, the *two moons* dataset consists of two banana-shaped distributions separable by a nonlinear decision boundary. We consider a 12-layer, 128-unit deep architecture ResFFN-12-128. The full experimental details are in Appendix C.

Figure 1 shows the results, where the background color visualizes the uncertainty surface output by each model. We first notice that the shallow Gaussian process models (Figures 1a and 1f) exhibit an expected behavior for high-quality predictive uncertainty: it generates low uncertainty in $\mathscr{X}_{\text{IND}}$ that is supported by the training data (purple color), and generates high uncertainty when $\mathbf{x}$ is far from $\mathscr{X}_{\text{IND}}$ (yellow color), i.e., *input distance awareness*. As a result, the shallow GP model is able to assign low confidence to the OOD data (colored in red), indicating reliable uncertainty quantification. On the other hand, deep ensembles (Figures 1b, 1g) and MC Dropout (Figures 1c, 1h) are based on dense output layers that are not distance aware. As a result, both methods quantify their predictive uncertainty based on the distance from the decision boundaries, assigning low uncertainty to OOD examples even if they are far from the data. Finally, the DNN-GP (Figures 1d and 1i) and SNGP (Figures 1e and 1j) both use GP as their output layers, but with SNGP additionally imposing the spectral normalization on its hidden mapping $h(.)$. As a result, the DNN-GP's uncertainty surfaces are still strongly impacted by the distance from decision boundary, likely caused by the fact that the un-regularized hidden mapping $h(\mathbf{x})$ is free to discard information that is not relevant for prediction. On the other hand, the SNGP is able to maintain the *input distance awareness* property via its

bi-Lipschitz constraint, and exhibits a uncertainty surface that is analogous to the gold-standard model (shallow GP) despite the fact that SNGP is based on a 12-layer network.

## 5.2 Vision and Language Understanding

**Baseline Methods** All methods included in the vision and language understanding experiments are summarized in Table 1. Specifically, we evaluate SNGP on a Wide ResNet 28-10 [83] for image classification, and BERT$_{\mathrm{base}}$ [18] for language understanding. We compare against a deterministic baseline and two ensemble approaches: **MC Dropout** (with 10 dropout samples) and **deep ensembles** (with 10 models), all trained with a dense output layer and no spectral regularization. We consider three single-model approaches: **MCD-GP** (with 10 samples), **Deterministic Uncertainty Quantification (DUQ)** (see Section 4). For all models that use GP layer, we keep $D_L = 1024$ and compute predictive distribution by performing Monte Carlo averaging with 10 samples. We also include two ablated version of SNGP: **DNN-SN** which uses spectral normalization on its hidden weights and a dense output layer (i.e. distance preserving hidden mapping without distance-aware output layer), and **DNN-GP** which uses the GP as output layer but without spectral normalization on its hidden layers (i.e., distance-aware output layer without distance-preserving hidden mapping). Further experiment details and recommendations for practical implementation are in Appendix C. All baselines are built on the `uncertainty_baselines` framework.

| Methods | Additional Regularization | Output Layer | Ensemble Training | Multi-pass Inference |
|---|---|---|---|---|
| Deterministic | - | Dense | - | - |
| MC Dropout | Dropout | Dense | - | Yes |
| Deep Ensemble | - | Dense | Yes | Yes |
| MCD-GP | Dropout | GP | - | Yes |
| DUQ | Gradient Penalty | RBF | - | - |
| DNN-SN | Spec Norm | Dense | - | - |
| DNN-GP | - | GP | - | - |
| SNGP | Spec Norm | GP | - | - |

Table 1: Summary of methods used in experiments. Multi-pass Inference refers to whether the method needs to perform multiple forward passes to generate the predictive distribution.

**CIFAR-10 and CIFAR-100** We evaluate the model's predictive accuracy and calibration error under both clean CIFAR testing data and its corrupted versions termed CIFAR-*-C [34]. To evaluate the model's OOD detection performance, we consider two tasks: a standard OOD task using SVHN as the OOD dataset for a model trained on CIFAR-10/-100, and a difficult OOD task using CIFAR-100 as the OOD dataset for a model trained on CIFAR-10, and vice versa. We compute the uncertainty score for OOD using the Dempster-Shafer metric as introduced in [68], which empirically leads to better performance for distance-aware models (see Appendix C). Table 2-3 reports the results. As shown, for predictive accuracy, SNGP is competitive with that of a deterministic network, and outperforms the other single-model approaches. For calibration error, SNGP clearly outperforms the other single-model approaches and is competitive with the deep ensemble. Finally, for OOD detection, SNGP outperforms not only the deep ensembles and MC Dropout approaches that are based on a dense output layer, but also the MCD-GP and DUQ that are based on the GP layer, illustrating the importance of the *input distance awareness* property for high-quality performance in uncertainty quantification.

| Method | Accuracy (↑) Clean | Corrupted | ECE (↓) Clean | Corrupted | NLL (↓) Clean | Corrupted | OOD AUPR (↑) SVHN | CIFAR-100 | Latency (↓) (ms / example) |
|---|---|---|---|---|---|---|---|---|---|
| Deterministic | $96.0 \pm 0.01$ | $72.9 \pm 0.01$ | $0.023 \pm 0.002$ | $0.153 \pm 0.011$ | $0.158 \pm 0.01$ | $1.059 \pm 0.02$ | $0.781 \pm 0.01$ | $0.835 \pm 0.01$ | **3.91** |
| MC Dropout | $96.0 \pm 0.01$ | $70.0 \pm 0.02$ | $0.021 \pm 0.002$ | $0.116 \pm 0.009$ | $0.173 \pm 0.01$ | $1.152 \pm 0.01$ | $0.971 \pm 0.01$ | $0.832 \pm 0.01$ | 27.10 |
| Deep Ensembles | $\mathbf{96.6 \pm 0.01}$ | $\mathbf{77.9 \pm 0.01}$ | $\mathbf{0.010 \pm 0.001}$ | $\mathbf{0.087 \pm 0.004}$ | $\mathbf{0.114 \pm 0.01}$ | $\mathbf{0.815 \pm 0.01}$ | $0.964 \pm 0.01$ | $\underline{0.888 \pm 0.01}$ | 38.10 |
| MCD-GP | $95.5 \pm 0.02$ | $70.0 \pm 0.01$ | $0.024 \pm 0.004$ | $0.100 \pm 0.007$ | $0.172 \pm 0.01$ | $1.157 \pm 0.01$ | $0.960 \pm 0.01$ | $0.863 \pm 0.01$ | 29.53 |
| DUQ | $94.7 \pm 0.02$ | $71.6 \pm 0.02$ | $0.034 \pm 0.004$ | $0.183 \pm 0.011$ | $0.239 \pm 0.02$ | $1.348 \pm 0.01$ | $0.973 \pm 0.01$ | $0.854 \pm 0.01$ | 8.68 |
| DNN-SN | $96.0 \pm 0.01$ | $72.5 \pm 0.01$ | $0.025 \pm 0.004$ | $0.178 \pm 0.013$ | $0.171 \pm 0.01$ | $1.306 \pm 0.01$ | $0.974 \pm 0.01$ | $0.859 \pm 0.01$ | 5.20 |
| DNN-GP | $\underline{95.9 \pm 0.01}$ | $71.7 \pm 0.01$ | $0.029 \pm 0.002$ | $0.175 \pm 0.008$ | $0.221 \pm 0.02$ | $1.380 \pm 0.01$ | $\underline{0.976 \pm 0.01}$ | $0.887 \pm 0.01$ | 5.58 |
| SNGP (Ours) | $\underline{95.9 \pm 0.01}$ | $74.6 \pm 0.01$ | $0.018 \pm 0.001$ | $\underline{0.090 \pm 0.012}$ | $0.138 \pm 0.01$ | $\underline{0.935 \pm 0.01}$ | $\mathbf{0.990 \pm 0.01}$ | $\mathbf{0.905 \pm 0.01}$ | 6.25 |

Table 2: Results for Wide ResNet-28-10 on CIFAR-10, averaged over 10 seeds.

**Detecting Out-of-Scope Intent in Conversational Language Understanding** To validate the method beyond image modalities, we also evaluate SNGP on a practical language understanding task where uncertainty quantification is of natural importance: dialog intent detection [44, 78, 82, 84]. In a goal-oriented dialog system (e.g. chatbot) built for a collection of in-domain services, it is important for the model to understand if an input natural utterance from an user is in-scope (so it can activate

| Method | Accuracy (↑) | | ECE (↓) | | NLL (↓) | | OOD AUPR (↑) | | Latency (↓) |
|---|---|---|---|---|---|---|---|---|---|
| | Clean | Corrupted | Clean | Corrupted | Clean | Corrupted | SVHN | CIFAR-10 | (ms / example) |
| Deterministic | 79.8 ± 0.02 | 50.5 ± 0.04 | 0.085 ± 0.004 | 0.239 ± 0.020 | 0.872 ± 0.01 | 2.756 ± 0.03 | 0.882 ± 0.01 | 0.745 ± 0.01 | **5.20** |
| MC Dropout | 79.6 ± 0.02 | 42.6 ± 0.08 | 0.050 ± 0.003 | 0.202 ± 0.010 | 0.825 ± 0.01 | 2.881 ± 0.01 | 0.832 ± 0.01 | 0.757 ± 0.01 | 46.79 |
| Deep Ensemble | **80.2 ± 0.01** | **54.1 ± 0.04** | **0.021 ± 0.004** | 0.138 ± 0.013 | **0.666 ± 0.02** | **2.281 ± 0.03** | 0.888 ± 0.01 | 0.780 ± 0.01 | 42.06 |
| MCD-GP | 79.5 ± 0.04 | 45.0 ± 0.05 | 0.085 ± 0.005 | 0.159 ± 0.009 | 0.937 ± 0.01 | 2.584 ± 0.02 | 0.873 ± 0.01 | 0.754 ± 0.01 | 44.20 |
| DUQ | 78.5 ± 0.02 | 50.4 ± 0.02 | 0.119 ± 0.001 | 0.281 ± 0.012 | 0.980 ± 0.02 | 2.841 ± 0.01 | 0.878 ± 0.01 | 0.732 ± 0.01 | 6.51 |
| DNN-SN | 79.9 ± 0.02 | 48.6 ± 0.02 | 0.098 ± 0.004 | 0.272 ± 0.011 | 0.918 ± 0.01 | 3.013 ± 0.01 | 0.879 ± 0.03 | 0.745 ± 0.01 | 6.20 |
| DNN-GP | 79.2 ± 0.03 | 47.7 ± 0.03 | 0.064 ± 0.005 | 0.166 ± 0.003 | 0.885 ± 0.009 | 2.629 ± 0.01 | 0.876 ± 0.01 | 0.746 ± 0.02 | 6.82 |
| SNGP (Ours) | 79.9 ± 0.03 | 49.0 ± 0.02 | 0.025 ± 0.012 | **0.117 ± 0.014** | 0.847 ± 0.01 | 2.626 ± 0.01 | **0.923 ± 0.01** | **0.801 ± 0.01** | 6.94 |

Table 3: Results for Wide ResNet-28-10 on CIFAR-100, averaged over 10 seeds.

one of the in-domain services) or out-of-scope (where the model should abstain). To this end, we consider training an intent understanding model using the CLINC OOS intent detection benchmark dataset [44]. Briefly, the OOS dataset contains data for 150 in-domain services with 150 training sentences in each domain, and also 1500 natural out-of-domain utterances. We train the models only on in-domain data, and evaluate their predictive accuracy on the in-domain test data, their calibration and OOD detection performance on the combined in-domain and out-of-domain data. The results are in Table 4. As shown, consistent with the previous vision experiments, SNGP is competitive in predictive accuracy when compared to a deterministic baseline, and outperforms other approaches in calibration and OOD detection.

| Method | Accuracy (↑) | ECE (↓) | NLL (↓) | OOD | | Latency (↓) |
|---|---|---|---|---|---|---|
| | | | | AUROC (↑) | AUPR (↑) | (ms / example) |
| Deterministic | 96.5 ± 0.11 | 0.024 ± 0.002 | 3.559 ± 0.11 | 0.897 ± 0.01 | 0.757 ± 0.02 | **10.42** |
| MC Dropout | 96.1 ± 0.10 | 0.021 ± 0.001 | 1.658 ± 0.05 | 0.938 ± 0.01 | 0.799 ± 0.01 | 85.62 |
| Deep Ensemble | **97.5 ± 0.03** | **0.013 ± 0.002** | **1.062 ± 0.02** | 0.964 ± 0.01 | 0.862 ± 0.01 | 84.46 |
| MCD-GP | 95.9 ± 0.05 | 0.015 ± 0.003 | 1.664 ± 0.04 | 0.906 ± 0.02 | 0.803 ± 0.01 | 88.38 |
| DUQ | 96.0 ± 0.04 | 0.059 ± 0.002 | 4.015 ± 0.08 | 0.917 ± 0.01 | 0.806 ± 0.01 | 15.60 |
| DNN-SN | 95.4 ± 0.10 | 0.037 ± 0.004 | 3.565 ± 0.03 | 0.922 ± 0.02 | 0.733 ± 0.01 | 17.36 |
| DNN-GP | 95.9 ± 0.07 | 0.075 ± 0.003 | 3.594 ± 0.02 | 0.941 ± 0.01 | 0.831 ± 0.01 | 18.93 |
| SNGP | 96.6 ± 0.05 | 0.014 ± 0.005 | 1.218 ± 0.03 | **0.969 ± 0.01** | **0.880 ± 0.01** | 17.36 |

Table 4: Results for BERT$_{\text{Base}}$ on CLINC OOS, averaged over 10 seeds.

## 6 Conclusion

We propose SNGP, a simple approach to improve a single deterministic DNN's ability in predictive uncertainty estimation. It makes minimal changes to the architecture and training/prediction pipeline of a deterministic DNN, only adding spectral normalization to the hidden mapping, and replacing the dense output layer with a random feature layer that approximates a GP. We theoretically motivate *input distance awareness*, the key design principle behind SNGP, via a learning-theoretic analysis of the uncertainty estimation problem. We also propose a closed-form approximation method to make the GP posterior end-to-end trainable in linear time with the rest of the neural network. On a suite of vision and language understanding tasks and on modern architectures (ResNet and BERT), SNGP is competitive with a deep ensemble in prediction, calibration and out-of-domain detection, and outperforms other single-model approaches.

A central observation we made in this work is that *good representational learning is important for good uncertainty quantification*. In particular, we highlighted *bi-Lipschitz* (Equation (5)) as an important condition for the learned representation of a DNN to attain high-quality uncertainty performance, and proposed spectral normalization as a simple approach to ensure such property in practice. However, it is worth noting that there exists other representation learning techniques, e.g., data augmentation or unsupervised pretraining, that are known to also improve a network's uncertainty performance [35, 36]. Analyzing whether and how these approaches contribute to improve a DNN *bi-Lipschitz* condition, and whether the *bi-Lipschitz* condition is sufficient in explaining these methods' success, are interesting avenues of future work. Furthermore, we note that the spectral norm bound $\alpha < 1$ in Proposition 1 forms only a sufficient condition for ensuring bi-Lipschitz [5]. In practice, we observed that for convolutional layers, a looser norm bound is needed for state-of-the-art performance (see Section C), raising questions of whether the current regularization approach is precise enough in controlling the spectral norm of a convolutional kernel, or if there is an alternative mechanism at play in ensuring the bi-Lipschitz criterion. Finally, from a probabilistic learning perspective, SNGP focuses on learning a single high-quality model $p_\theta(y|\mathbf{x})$ for a deterministic representation. Therefore we expect it to provide complementary benefits to approaches such as (efficient) ensembles and Bayesian neural networks [21, 42, 79] which marginalize over the representation parameters as well.

**Acknowledgements** We would like to thank Kevin Murphy, Deepak Ramachandran, Jasper Snoek, and Timothy Nguyen at Google Research for the insightful comments and fruitful discussion.

## Broader Impact

This work proposed a simple and practical methodology to improve the uncertainty estimation performance of a deterministic deep learning model. Experiment results showcased the method's ability in improving model performance in calibration and OOD detection while maintaining similar level of accuracy and latency, therefore illustrating its feasibility for industrial-scale applications. We hope the proposed approach can be used to bring concrete improvements to AI-driven, socially-relevant services where uncertainty is of natural importance. Examples include medical and policy decision making, online toxic comment management, fairness-aware recommendation systems, etc.

Nonetheless, we do not claim that the improvement illustrated in this paper solve the problem of model uncertainty entirely. This is because the analysis and experiments in this study may not capture the full complexity of the real-world use cases, and there will always be room for improvement. Designers of machine learning systems are encouraged to proactively confront the shortcomings of model uncertainty and the underlying models that generate these confidences. Even with a proper user interface, there is always room to misinterpret model outputs and probabilities, such as with nuanced applications such as election predictions, and users of these models should to be properly trained to take these factors into account.

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
