[Supplementary Material]

# Supplementary Material:
# Simple and Principled Uncertainty Estimation with Deterministic Deep Learning via Distance Awareness

## A  Method Summary

**Architecture**  Given a deep learning model $\text{logit}(\mathbf{x}) = g \circ h(\mathbf{x})$ with $L-1$ hidden layers of size $\{D_l\}_{l=1}^{L}$, SNGP makes two changes to the model:

1. adding spectral normalization to the hidden weights $\{\mathbf{W}_l\}_{l=1}^{L}$, and

2. replacing the dense output layer $g(h) = h^{\top}\beta$ with a GP layer. Under the RFF approximation, the GP layer is simply a one-layer network with $D_L$ hidden units $g(h) \propto \cos(-\mathbf{W}_L h_i + \mathbf{b}_L)^{\top}\beta$, where $\{\mathbf{W}_L, \mathbf{b}_L\}$ are frozen weights that are initialized from a Gaussian and a uniform distribution, respectively (as described in Equation (8)).

**Training**  Algorithm 1 summarizes the training step. As shown, for every minibatch step, the model first updates the hidden-layer weights $\{\mathbf{W}_l, \mathbf{b}_l\}_{l=1}^{L-1}$ and the trainable output weights $\beta = \{\beta_k\}_{k=1}^{K}$ via SGD, then performs spectral normalization using power iteration method ([26] which has time complexity $O(\sum_{l=1}^{L-1} D_l)$), and finally performs precision matrix update (Equation (9), time complexity $O(D_L^2)$). Since $\{D_l\}_{l=1}^{L-1}$ are fixed for a given architecture and usually $D_L \leq 1024$, the computation scales linearly with respect to the sample size. We use $D_L = 1024$ in the experiments.

**Prediction**  Algorithm 2 summarizes the prediction step. The model first performs the conventional forward pass, which involves (i) computing the final hidden feature $\Phi(\mathbf{x})_{D_L \times 1}$, and (ii) computing the posterior mean $\hat{m}_k(\mathbf{x}) = \Phi^{\top}\beta_k$ (time complexity $O(D_L)$). Then, using the posterior variance matrices $\{\hat{\Sigma}_k\}_{k=1}^{K}$, the model computes the predictive variance $\hat{\sigma}_k(\mathbf{x})^2 = \Phi(\mathbf{x})^{\top}\hat{\Sigma}\Phi(\mathbf{x})$ (time complexity $O(D_L^2)$).

To estimate the predictive distribution $p_k = \exp(m_k)/\sum_k \exp(m_k)$ where $m_k \sim N\big(\hat{m}_k(\mathbf{x}), \hat{\sigma}_k^2(\mathbf{x})\big)$, we calculate its posterior mean using Monte Carlo averaging. Notice that this Monte Carlo averaging is computationally very cheap since it only involves sampling from a closed-form distribution whose parameters $(\hat{m}, \hat{\sigma}^2)$ are already computed by the single feed-forward pass (i.e., a single call to `tf.random.normal`). This is different from the full Monte Carlo sampling used by MC Dropout or deep ensembles which require multiple forward passes and are computationally expensive. As shown in the latency results in experiments (Section 5.2), the extra variance-related computation only adds a small overhead to the inference time of a deterministic DNN. In the experiments, we use 10 samples to compute the mean predictive distribution.

In applications where the inference latency is of high priority (e.g., real-time pCTR prediction for online advertising), we can reduce the computational overhead further by replacing the Monte Carlo averaging with the mean-field approximation [15]. We leave this for future work.

# B  Formal Statements

**Minimax Solution to Uncertainty Risk Minimization**   The expression in (4) seeks to answer the following question: *assuming we know the true domain probability $p^*(\mathbf{x} \in \mathcal{X}_{IND})$, and given a model $p(y|\mathbf{x}, \mathbf{x} \in \mathcal{X}_{IND})$ that we have already learned from data, what is the best solution we can construct to minimize the minimax objective (3)?* The interest of this conclusion is not to construct a practical algorithm, but to highlight the theoretical necessity of taking into account the domain probability in constructing a good solution for uncertainty quantification. If the domain probability is not necessary, then the expression of the unique and optimal solution to the minimax probability should not contain $p^*(\mathbf{x} \in \mathcal{X}_{IND})$ even if it is available. However the expression of (4) shows this is not the case.

To make the presentation clear, we formalize the statement about (4) into the below proposition:

**Proposition 2** (Minimax Solution to Uncertainty Risk Minimization).
*Given:*

(a) $p(y|\mathbf{x}, \mathbf{x} \in \mathcal{X}_{IND})$ *the model's predictive distribution learned from data* $\mathcal{D} = \{y_i, \mathbf{x}_i\}_{i=1}^N$

(b) $p^*(\mathbf{x} \in \mathcal{X}_{IND})$ *the true domain probability,*

*then there exists an unique optimal solution to the minimax problem (3), and it can be constructed using (a) and (b) as:*

$$p(y|\mathbf{x}) = p(y|\mathbf{x}, \mathbf{x} \in \mathcal{X}_{IND}) * p^*(\mathbf{x} \in \mathcal{X}_{IND}) + p_{\text{uniform}}(y|\mathbf{x}, \mathbf{x} \notin \mathcal{X}_{IND}) * p^*(\mathbf{x} \notin \mathcal{X}_{IND}) \tag{11}$$

*where $p_{\text{uniform}}(y|\mathbf{x}, \mathbf{x} \notin \mathcal{X}_{IND}) = \frac{1}{K}$ is a discrete uniform distribution for K classes.*

As discussed in Section 2.1, the solution (11) is not only optimal for the minimax Brier risk, but is in fact optimal for a wide family of strictly proper scoring rules known as the (separable) *Bregman score* [58]:

$$s(p, p^*|\mathbf{x}) = \sum_{k=1}^{K} \left\{ [p^*(y_k|\mathbf{x}) - p(y_k|\mathbf{x})] \, \psi'(p^*(y_k|\mathbf{x})) - \psi(p^*(y_k|\mathbf{x})) \right\} \tag{12}$$

where $\psi$ is a strictly concave and differentiable function. Bregman score reduces to the log score when $\psi(p) = p * \log(p)$, and reduces to the Brier score when $\psi(p) = p^2 - \frac{1}{K}$.

Therefore we will show (11) for the Bregman score. The proof relies on the following key lemma:

**Lemma 1** ($p_{\text{uniform}}$ is Optimal for Minimax Bregman Score in $\mathbf{x} \notin \mathcal{X}_{IND}$).
*Consider the Bregman score in (12). At a location $\mathbf{x} \notin \mathcal{X}_{IND}$ where the model has no information about $p^*$ other than $\sum_{k=1}^{K} p(y_k|\mathbf{x}) = 1$, the solution to the minimax problem*

$$\inf_{p \in \mathscr{P}} \sup_{p^* \in \mathscr{P}^*} s(p, p^*|\mathbf{x})$$

*is the discrete uniform distribution, i.e., $p_{uniform}(y_k|\mathbf{x}) = \frac{1}{K} \forall k \in \{1, \ldots, K\}$.*

The proof for Lemma 1 is in Section E.2. It is worth noting that Lemma 1 only holds for a *strictly proper scoring rule* [24]. For a non-strict proper scoring rule (e.g., the ECE), there can exist infinitely many optimal solutions, making the minimax problem ill-posed.

We are now ready to prove Proposition 2:

*Proof.* Denote $\mathcal{X}_{OOD} = \mathcal{X}/\mathcal{X}_{IND}$. Decompose the overall Bregman risk by domain:

$$
\begin{aligned}
S(p, p^*) &= E_{x \in \mathscr{X}}\left(s(p, p^*|\mathbf{x})\right) = \int_{\mathscr{X}} s(p, p^*|\mathbf{x}) p^*(\mathbf{x}) d\mathbf{x} \\
&= \int_{\mathscr{X}} s(p, p^*|\mathbf{x}) * \left[ p^*(\mathbf{x}|\mathbf{x} \in \mathcal{X}_{IND}) p^*(\mathbf{x} \in \mathcal{X}_{IND}) + p^*(\mathbf{x}|\mathbf{x} \in \mathcal{X}_{OOD}) p^*(\mathbf{x} \in \mathcal{X}_{OOD}) \right] d\mathbf{x} \\
&= E_{\mathbf{x} \in \mathcal{X}_{IND}}\left(s(p, p^*|\mathbf{x})\right) p^*(\mathbf{x} \in \mathcal{X}_{IND}) + E_{\mathbf{x} \in \mathcal{X}_{OOD}}\left(s(p, p^*|\mathbf{x})\right) p^*(\mathbf{x} \in \mathcal{X}_{OOD}) \\
&= S_{IND}(p, p^*) * p^*(\mathbf{x} \in \mathcal{X}_{IND}) + S_{OOD}(p, p^*) * p^*(\mathbf{x} \in \mathcal{X}_{OOD}).
\end{aligned}
$$

where we have denoted $S_{IND}(p, p^*) = E_{\mathbf{x} \in \mathcal{X}_{IND}}\left(s(p, p^*|\mathbf{x})\right)$ and $S_{OOD}(p, p^*) = E_{\mathbf{x} \in \mathcal{X}_{OOD}}\left(s(p, p^*|\mathbf{x})\right)$.

Now consider decomposing the sup risk $\sup_{p^*} S(p, p^*)$ for a given $p$. Notice that sup risk $\sup_{p^*} S(p, p^*)$ is separable by domain for any $p \in \mathscr{P}$. This is because $S_{\text{IND}}(p, p^*)$ and $S_{\text{OOD}}(p, p^*)$ has disjoint support, and we do not impose assumption on $p^*$:

$$\sup_{p^*} S(p, p^*) = \sup_{p^*} \left[ S_{\text{IND}}(p, p^*) \right] * p^*(\mathbf{x} \in \mathscr{X}_{\text{IND}}) + \sup_{p^*} \left[ S_{\text{OOD}}(p, p^*) \right] * p^*(\mathbf{x} \in \mathscr{X}_{\text{OOD}})$$

We are now ready to decompose the minimax risk $\inf_p \sup_{p^*} S(p, p^*)$. Notice that the minimax risk is also separable by domain due to the disjoint in support:

$$\inf_p \sup_{p^*} S(p, p^*) = \inf_p \left[ \sup_{p^*} \left[ S_{\text{IND}}(p, p^*) \right] * p^*(\mathbf{x} \in \mathscr{X}_{\text{IND}}) + \sup_{p^*} \left[ S_{\text{OOD}}(p, p^*) \right] * p^*(\mathbf{x} \in \mathscr{X}_{\text{OOD}}) \right]$$

$$= \inf_p \sup_{p^*} \left[ S_{\text{IND}}(p, p^*) \right] * p^*(\mathbf{x} \in \mathscr{X}_{\text{IND}}) + \inf_p \sup_{p^*} \left[ S_{\text{OOD}}(p, p^*) \right] * p^*(\mathbf{x} \in \mathscr{X}_{\text{OOD}}), \quad (13)$$

also notice that the in-domain minimax risk $\inf_p \sup_{p^*} \left[ S_{\text{IND}}(p, p^*) \right]$ is fixed due to condition $(a)$.

Therefore, to show that (11) is the optimal and unique solution to (13), we only need to show $p_{uniform}$ is the optimal and unique solution to $\inf_p \sup_{p^*} \left[ S_{\text{OOD}}(p, p^*) \right]$. To this end, notice that for a given $p$:

$$\sup_{p^* \in \mathscr{P}^*} \left[ S_{\text{OOD}}(p, p^*) \right] = \int_{\mathscr{X}_{\text{OOD}}} \sup_{p^*} [s(p, p^*|\mathbf{x})] p(\mathbf{x}|\mathbf{x} \in \mathscr{X}_{\text{OOD}}) d\mathbf{x}, \quad (14)$$

due to the fact that we don't impose assumption on $p^*$ (therefore $p^*$ is free to attain the global supreme by maximizing $s(p, p^*|\mathbf{x})$ at every single location $\mathbf{x} \in \mathscr{X}_{\text{OOD}}$). Furthermore, there exists $p$ that minimize $\sup_{p^*} s(p, p^*|\mathbf{x})$ at every location of $\mathbf{x} \in \mathscr{X}_{\text{OOD}}$, then it minimizes the integral [7]. By Lemma 1, such $p$ exists and is unique, i.e.:

$$p_{uniform} = \arginf_{p \in \mathscr{P}} \sup_{p^* \in \mathscr{P}^*} S_{\text{OOD}}(p, p^*).$$

In conclusion, we have shown that $p_{uniform}$ is the unique solution to $\inf_p \sup_{p^*} S_{\text{OOD}}(p, p^*)$. Combining with condition (a)-(b), we have shown that the unique solution to (13) is (11). □

## C    Experiment Details and Further Results

### C.1    2D Synthetic Benchmark

For both benchmarks, we sample 500 observations $\mathbf{x}_i = (x_{1i}, x_{2i})$ from each of the two in-domain classes (orange and blue), and consider a deep architecture ResFFN-12-128, which contains 12 residual feedforward layers with 128 hidden units and dropout rate 0.01. The input dimension is projected from 2 dimensions to the 128 dimensions using a dense layer.

In addition to SNGP, we also visualize the uncertainty surface of the below approaches: **Gaussian process (GP)** is a standard Gaussian process directly taking $\mathbf{x}_i$ as in input. In low-dimensional datasets, GP is often considered the gold standard for uncertainty quantification. **Deep Ensemble** is an ensemble of 10 ResFFN-12-128 models with dense output layers, **MC Dropout** uses single ResFFN-12-128 model with dense output layer and 10 dropout samples. **DNN-GP** uses a single ResFFN-12-128 model with the GP Layer (described in Section 3.1) without spectral normalization. Finally, **SNGP** uses a single ResFFN-12-128 model with the GP layer and with the spectral normalization.

For these two binary classification tasks, in Figure 1 we plot the predictive uncertainty for **GP**, **DNN-GP** and **SNGP** as the posterior predictive variance of the logits, i.e., $u(\mathbf{x}) = var(logit(\mathbf{x}))$ which ranges between $[0, 1]$ under the RBF kernel. For **MC Dropout** and **Deep Ensemble**, since these two methods don't provide an convenient expression of predictive variance, we plot their predictive uncertainty as the distance of the maximum predictive probability from 0.5, i.e., $u(\mathbf{x}) = 1 - 2 * |p(\mathbf{x}) - 0.5|$, so that $u(\mathbf{x}) \in [0, 1]$.

Figure 2-3 compares the aforementioned methods in terms of the same metric based on the predictive probability introduced in the last paragraph: $u(\mathbf{x}) = 1 - 2 * |p - 0.5|$. We also included **DNN-SN** (a ResFFN-12-128 model with spectral normalization but no GP layer) into comparison. As shown, compared to the uncertainty surface based on predictive variance (Figure 1), the uncertainty surface based on predictive probability shows stronger influence from the model's decision boundary. This empirical observation seems to suggest that the predictive uncertainty from the GP logits can be a better metric for calibration and OOD detection. We will explore the performance difference of different uncertainty metrics in calibration and OOD performance in the future work.

Figure 2: The uncertainty surface of a GP and different DNN approaches on the *two ovals* 2D classification benchmarks. The uncertainty is computed in terms of the distance of the maximum predictive probability from 0.5, i.e. $u(\mathbf{x}) = 1 - 2 * |p(\mathbf{x}) - 0.5|$. Background color represents the estimated model uncertainty (See 1e for color map).

Figure 3: The uncertainty surface of a GP and different DNN approaches on the *two moons* 2D classification benchmarks. The uncertainty is computed in terms of the distance of the maximum predictive probability from 0.5, i.e. $u(\mathbf{x}) = 1 - 2 * |p(\mathbf{x}) - 0.5|$. Background color represents the estimated model uncertainty (See Figures 1j and 1e for color map).

## C.2 Vision and Language Understanding

**Hyperparameter Configuration** SNGP is composed of two components: Spectral Normalization (SN) and Gaussian Process (GP) layer, both are available at the open-source Edward2 probabilistic programming library [4].

Spectral normalization contains two hyperparameters: the number of power iterations and the upper bound for spectral norm (i.e., $c$ in Equation (10)). In our experiments, we find it is sufficient to fix power iteration to 1. The value for the spectral norm bound $c$ controls the trade-off between the expressiveness and the distance-awareness of the residual block, where a small value of $c$ may shrink the residual block back to identity mapping hence harming the expressiveness, while a large value of $c$ may lead to the loss of bi-Lipschitz property (Proposition 1). Furthermore, the proper range of $c$ depends on the layer type: for dense layers (e.g., the intermediate and the output dense layers of a Transformer), it is sufficient to set $c$ to a value between $(0.95, 1)$. For the convolutional layers, the norm bound needs to be set to a larger value to not impact the model's predictive performance. This is likely caused by the fact that the current spectral normalization technique does not have a precise control of the true spectral norm of the convolutional kernel, in conjuction with the fact that the other regularization mechanisms (e.g., BatchNorm and Dropout) may rescale a layer's spectral norm in unexpected ways [26, 54]. In general, we recommend performing a grid search for $c \in \{0.9, 1, 2, ...\}$ to identify the smallest possible values of $c$ that still retains the predictive performance of the original model. In the experiments, we set the norm bound to $c = 6$ for a WideResNet model.

The GP layer contains 3 hyperparameters for the main layer (Equation (8)), and 2 hyperparameters for its covariance module (Equation (9). The three hyperparameter for the main layers are the *hidden dimension* ($D_L$, i.e., the number of random features), the *length-scale parameter l* for the RBF kernel, and the strength of $L_2$ regularization on output weights $\beta_k$. In the experiments, we find the model's performance to be not very sensitive to these parameters. Setting $D_L = 1024$ or $2048$, $l = 2.0$ and $L_2$ regularization to 0 are sufficient in most cases. The two hyperparameters for the covariance module is the ridge factor $s$ and the discount factor $m$, they come into the update rule of the precision matrix as:

$$\Sigma_{k,0}^{-1} = s * \mathbf{I}, \quad \Sigma_{k,t}^{-1} = m * \Sigma_{k,t-1}^{-1} + (1-m) * \sum_{i=1}^{M} \hat{p}_{ik}(1 - \hat{p}_{ik})\Phi_i\Phi_i^{\top},$$

i.e., the ridge factor $s$ serves to control the stability of matrix inverse (if the number of sample size $n$ is small), and $m$ controls how fast the moving average update converges to the population value $\Sigma_k = s\mathbf{I} + \sum_{i=1}^{n} \hat{p}_{ik}(1 - \hat{p}_{ik})\Phi_i\Phi_i^{\top}$. Similar to other moving-average update method, these two parameters can impact the quality of learned covariance matrix in non-trivial ways. In general, we recommend conducting some small scale experiments on the data to validate the learning quality of the moving average update in approximating the population covariance. Alternatively, the covariance update can be computed exactly at the final epoch by initialize $\Sigma_{k,0}^{-1} = \mathbf{0}$ and simply using the update formula $\Sigma_{k,t}^{-1} = \Sigma_{k,t-1}^{-1} + \sum_{i=1}^{M} \hat{p}_{ik}(1 - \hat{p}_{ik})\Phi_i\Phi_i^{\top}$. In the experiments, we set $s = 0.001$ and $m = 0.999$, which is sufficient for our setting where the number of minibatch steps per epoch is large.

We also implemented two additional functionalities for GP layers: *input dimension projection* and *input layer normalization*. The input dimension project serves to project the hidden dimension of the penultimate layer $D_{L-1}$ to a lower value $D'_{L-1}$ (using a random Gaussian matrix $\mathbf{W}_{D_{L-1} \times D'_{L-1}}$), it can be projected down to a smaller dimension. *Input layer normalization* applies Layer Normalization to the input hidden features, which is akin to performing automatic relevance determination (ARD)-style variable selection to the input features. In the experiments, we always turn on the *input layer normalization* and set input layer normalization to 128. Although later ablation studies revealed that the model performance is not sensitive to these values.

**Data Preparation and Computing Infrastructure** For CIFAR-10 and CIFAR-100, we followed the original Wide ResNet work to apply the standard data augmentation (horizontal flips and random crop-ping with 4x4 padding) and used the same hyperparameter and training setup [83]. The only exception is the learning rate and training epochs, where we find a smaller learning rate (0.04 for CIFAR-10 and 0.08 for CIFAR100, v.s. 0.1 for the original WRN model) and longer epochs (250 for SNGP v.s. 200 for the original WRN model) leads to better performance.

| Spectral Normalization | | Gaussian Process Layer | |
|---|---|---|---|
| Power Iteration | 1 | Hidden Dimension | 1024 (WRN), 2048 (BERT) |
| **Spectral Norm Bound** | 6.0 (WRN), 0.95 (BERT) | Length-scale Parameter | 2.0 |
| | | $L_2$ Regularization | 0.0 |
| | | **Covariance Ridge Factor** | 0.001 |
| | | **Covariance Discount Factor** | 0.999 |
| | | Projected Input Dimension | 128 (WRN) None (BERT) |
| | | Input Layer Normalization | True |

Table 5: Hyperparameters of SNGP used in the experiments, where important hyperparameters are highlighted in bold.

For CLINC OOS intent understanding data, we pre-tokenized the sentences using the standard BERT tokenizer[5] with maximum sequence length 32, and created standard binary input mask for the BERT model that returns 1 for valid tokens and 0 otherwise. Following the original BERT work, we used the Adam optimizer with weight decay rate 0.01 and warmup proportion 0.1. We initialize the model from the official BERT$_{\texttt{Base}}$ checkpoint[6]. For this fine-tuning task, we using a smaller step size ($5e-5$ for SNGP .v.s. $1e-4$ for the original BERT model) but shorter epochs (40 for SNGP v.s. 150 for the original BERT model) leads to better performance. When using spectral normalization, we set the hyperparameter $c = 0.95$ and apply it to the pooler dense layer of the classification token. We do not spectral normalization to the hidden transformer layers, as we find the pre-trained BERT representation is already competent in preserving input distance due to the masked language modeling training, and further regularization may in fact harm its predictive and calibration performance.

All models are implemented in TensorFlow and are trained on 8-core Cloud TPU v2 with 8 GiB of high-bandwidth memory (HBM) for each TPU core. We use batch size 32 per core.

**Evaluation** For CIFAR-10 and CIFAR-100, we evaluate the model's predictive accuracy and calibration error under both clean and corrupted versions of the CIFAR testing data. The corrupted data, termed CIFAR10-C, includes 15 types of corruptions, e.g., noise, blurring, pixelation, etc, over 5 levels of corruption intensity [34]. We also evaluate the model performance in OOD detection by using the CIFAR-10/CIFAR-100 model's uncertainty estimate as a predictive score for OOD classification, where we consider a standard OOD task by testing CIFAR-10/CIFAR-100 model's ability in detecting samples from the Street View House Numbers (SVHN) dataset [55], and a more difficult OOD task by testing CIFAR-10's ability in detecting samples from the CIFAR-100 dataset, and vice versa. Specifically, for all models, we compute the OOD uncertainty score using the so-called *Dempster-Shafer metric* [68], which empirically leads to better performance for a distance-aware model. Given logits for $K$ classes $\{h_k(\mathbf{x}_i)\}_{k=1}^K$, this metric computes its uncertainty for a test example $\mathbf{x}_i$ as:

$$u(\mathbf{x}_i) = \frac{K}{K + \sum_{k=1}^K \exp\left(h_k(\mathbf{x}_i)\right)}. \tag{15}$$

As shown, for a distance-aware model where the magnitude of the logits reflects the distance from the observed data manifold, $u(\mathbf{x}_i)$ can be a more effective metric since it is monotonic to the magnitude of the logits. On the other hand, the maximum probability $p_{\texttt{max}} = \arg\max_k \exp(h_k)/\sum_{k=1}^K \exp\left(h_k(\mathbf{x}_i)\right)$ does not take advantage of this information since it normalizes over the exponentiated logits.

In terms of evaluation metrics, we assess the model's calibration performance using the empirical estimate of ECE: $\hat{ECE} = \sum_{m=1}^M \frac{|B_m|}{n} |acc(B_m) - conf(B_m)|$ which estimates the difference in model's accuracy and confidence by partitioning model prediction into $M$ bins $\{B_m\}_{m=1}^M$ [29]. In this work, we choose $M = 15$. We assess the model's OOD performance using Area Under Precision-Recall (AUPR). Finally, we measure each method's inference latency by millisecond per image.

For CLINC OOS intent detection data, we evaluate the predictive accuracy on the in-domain test data, evaluate the ECE and OOD detection performance on the combined in-domain and out-of-domain testing data, and we measure inference latency by millisecond per sentence.

# D   Additional Related Work

**Distance-preserving neural networks and bi-Lipschitz condition** The theoretic connection between distance preservation and the bi-Lipschitz condition is well-established [67], and learning an

approximately isometric, distance-preserving transform has been an important goal in the fields of dimensionality reduction [8, 59], generative modeling [45, 19, 20, 39], and adversarial robustness [37, 65, 71, 75, 80]. This work is a novel application of the distance preservation property for uncertainty quantification. There existing several methods for controlling the Lipschitz constant of a DNN (e.g., gradient penalty or norm-preserving activation [1, 2, 13, 28]), and we chose spectral normalization in this work due to its simplicity and its minimal impact on a DNN's architecture and the optimization dynamics [3, 5, 64].

**Open Set Classification** The uncertainty risk minimization problem in Section 2 assumes a data-generation mechanism similar to the *open set recognition* problem [66], where the whole input space is partitioned into known and unknown domains. However, our analysis is unique in that it focuses on measuring a model's behavior in uncertainty quantification and takes a rigorous, decision-theoretic approach to the problem. As a result, our analysis works with a special family of risk functions (i.e., *the strictly proper scoring rule*) that measure a model's performance in uncertainty calibration. Furthermore, it handles the existence of unknown domain via a minimax formulation, and derives the solution by using a generalized version of maximum entropy theorem for the Bregman scores [27, 43]. The form of the optimal solution we derived in (4) takes an intuitive form, and has been used by many empirical work as a training objective to leverage adversarial training and generative modeling to detect OOD examples [30, 31, 47, 51, 52]. Our analysis provide strong theoretical support for these practices in verifying rigorously the uniqueness and optimality of this solution, and also provides a conceptual unification of the notion of calibration and the notion of OOD generalization. Furthermore, it is used in this work to motivate a design principle (*input distance awareness*) that enables strong OOD performance in discriminative classifiers without the need of explicit generative modeling.

# E Proof

## E.1 Proof of Proposition 1

The proof for Proposition 1 is an adaptation of the classic result of [3] to our current context:

*Proof.* First establish some notations. We denote $I(\mathbf{x}) = \mathbf{x}$ the identity function such that for $h(\mathbf{x}) = \mathbf{x} + g(\mathbf{x})$, we can write $g = h - I$. For $h : \mathscr{X} \to \mathscr{H}$, denote $||h|| = \sup \left\{ \frac{||f(\mathbf{x})||_H}{||\mathbf{x}||_X} \text{ for } \mathbf{x} \in \mathscr{X}, ||\mathbf{x}|| > 0 \right\}$. Also denote the Lipschitz seminorm for a function $h$ as:

$$||h||_L = \sup \left\{ \frac{||h(\mathbf{x}) - h(\mathbf{x}')||_H}{||\mathbf{x} - \mathbf{x}'||_X} \quad \text{for} \quad \mathbf{x}, \mathbf{x}' \in \mathscr{X}, \mathbf{x} \neq \mathbf{x}' \right\} \tag{16}$$

It is worth noting that by the above definitions, for two functions $(\mathbf{x}' - \mathbf{x}) : \mathscr{X} \times \mathscr{X} \to \mathscr{X}$ and $(h(\mathbf{x}) - h(\mathbf{x}')) : \mathscr{X} \times \mathscr{X} \to \mathscr{H}$ who shares the same input space, the Lipschitz inequality can be expressed using the $||.||$ norm, i.e., $||h(\mathbf{x}) - h(\mathbf{x}')||_H \leq \alpha ||\mathbf{x} - \mathbf{x}'||_X$ implies $||h(\mathbf{x}') - h(\mathbf{x})|| \leq \alpha ||\mathbf{x} - \mathbf{x}'||$, and vice versa.

Now assume $\forall l, ||g_l||_L = ||h_l - I||_L \leq \alpha < 1$. We will show Proposition 1 by first showing:

$$(1 - \alpha)||\mathbf{x} - \mathbf{x}'|| \leq ||h_l(\mathbf{x}) - h_l(\mathbf{x}')|| \leq (1 + \alpha)||\mathbf{x} - \mathbf{x}'||, \tag{17}$$

which is the bi-Lipschitz condition for a single residual block.

First show the left hand side:

$$\begin{aligned} ||\mathbf{x} - \mathbf{x}'|| &\leq ||\mathbf{x} - \mathbf{x}' - (h_l(\mathbf{x}) - h_l(\mathbf{x}')) + (h_l(\mathbf{x}) - h_l(\mathbf{x}'))|| \\ &\leq ||(h_l(\mathbf{x}') - \mathbf{x}') - (h_l(\mathbf{x}) - \mathbf{x})|| + ||h_l(\mathbf{x}) - h_l(\mathbf{x}')|| \\ &\leq ||g_l(\mathbf{x}') - g_l(\mathbf{x})|| + ||h_l(\mathbf{x}) - h_l(\mathbf{x}')|| \\ &\leq \alpha ||\mathbf{x}' - \mathbf{x}|| + ||h_l(\mathbf{x}) - h_l(\mathbf{x}')||, \end{aligned}$$

where the last line follows by the assumption $||g_l||_L \leq \alpha$. Rearranging, we get:

$$(1 - \alpha)||\mathbf{x} - \mathbf{x}'|| \leq ||h_l(\mathbf{x}) - h_l(\mathbf{x}')||. \tag{18}$$

Now show the right hand side:

$$||h_l(\mathbf{x}) - h_l(\mathbf{x}')|| = ||\mathbf{x} + g_l(\mathbf{x}) - (\mathbf{x}' + g_l(\mathbf{x}'))|| \leq ||\mathbf{x} - \mathbf{x}'|| + ||g_l(\mathbf{x}) - g_l(\mathbf{x}'))|| \leq (1 + \alpha)||\mathbf{x} - \mathbf{x}'||.$$

Combining (18)-(19), we have shown (17), which also implies:

$$(1-\alpha)||\mathbf{x}-\mathbf{x}'||_X \leq ||h_l(\mathbf{x})-h_l(\mathbf{x}')||_H \leq (1+\alpha)||\mathbf{x}-\mathbf{x}'||_X \tag{19}$$

Now show the bi-Lipschitz condition for a $L$-layer residual network $h = h_L \circ h_{L-1} \circ \cdots \circ h_1$. It is easy to see that by induction:

$$(1-\alpha)^L||\mathbf{x}-\mathbf{x}'||_X \leq ||h(\mathbf{x})-h(\mathbf{x}')||_H \leq (1+\alpha)^L||\mathbf{x}-\mathbf{x}'||_X \tag{20}$$

Denoting $L_1 = (1-\alpha)^L$ and $L_2 = (1+\alpha)^L$, we have arrived at expression in Proposition 1. $\square$

### E.2   Proof of Lemma 1

*Proof.* This proof is an application of the generalized maximum entropy theorem to the case of Bregman score. We shall first state the generalized maximum entropy theorem to make sure the proof is self-contained. Briefly, the generalized maximum entropy theorem verifies that for a general scoring function $s(p, p^*|\mathbf{x})$ with entropy function $H(p|\mathbf{x})$, the maximum-entropy distribution $p' = \underset{p}{argsup}\, H(p|\mathbf{x})$ attains the minimax optimality :

**Theorem 1** (Maximum Entropy Theorem for General Loss [27]). *Let $\mathscr{P}$ be a convex, weakly closed and tight set of distributions. Consider a general score function $s(p, p^*|\mathbf{x})$ with an associated entropy function defined as $H(p|\mathbf{x}) = \inf_{p^* \in \mathscr{P}^*} s(p, p^*|\mathbf{x})$. Assume below conditions on $H(p|\mathbf{x})$ hold:*

- *(Well-defined) For any $p \in \mathscr{P}$, $H(p|\mathbf{x})$ exists and is finite.*

- *(Lower-semicontinous) For a weakly converging sequence $p_n \to p_0 \in \mathscr{P}$ where $H(p_n|\mathbf{x})$ is bounded below, we have $s(p, p_0|\mathbf{x}) \leq \liminf_{n \to \infty} s(p, p_n|\mathbf{x})$ for all $p \in \mathscr{P}$.*

*Then there exists an maximum-entropy distribution $p'$ such that*

$$p' = \sup_{p \in \mathscr{P}} H(p) = \sup_{p \in \mathscr{P}} \inf_{p^* \in \mathscr{P}^*} s(p, p^*|\mathbf{x}) = \inf_{p \in \mathscr{P}} \sup_{p^* \in \mathscr{P}^*} s(p, p^*|\mathbf{x}).$$

Above theorem states that the maximum-entropy distribution attains the minimax optimality for a scoring function $s(p, p^*|\mathbf{x})$, assuming its entropy function satisfying certain regularity conditions. Authors of [27] showed that the entropy function of a Bregman score satisfies conditions in Theorem 1. Consequently, to show that the discrete uniform distribution is minimax optimal for Bregman score at $\mathbf{x} \notin \mathscr{X}_{\texttt{IND}}$, we only need to show discrete uniform distribution is the maximum-entropy distribution.

Recall the definition of the *strictly* proper Bregman score [58]:

$$s(p, p^*|\mathbf{x}) = \sum_{k=1}^{K} \left\{ [p^*(y_k|\mathbf{x}) - p(y_k|\mathbf{x})] \psi'(p^*(y_k|\mathbf{x})) - \psi(p^*(y_k|\mathbf{x})) \right\} \tag{21}$$

where $\psi$ is differentiable and *strictly* concave. Moreover, its entropy function is:

$$H(p|\mathbf{x}) = -\sum_{k=1}^{K} \psi(p(y_k|\mathbf{x})) \tag{22}$$

Our interest is to show that for $\mathbf{x} \in \mathscr{X}_{\texttt{OOD}}$, the maximum-entropy distribution for the Bregman score is the discrete uniform distribution $p(y_k|\mathbf{x}) = \frac{1}{K}$. To this end, we notice that in the absence of any information, the only constraint on the predictive distribution is that $\sum_k p(y_k|\mathbf{x}) = 1$. Therefore, denoting $p(y_k|\mathbf{x}) = p_k$, we can set up the optimization problem with respect to Bregman entropy (22) using the Langrangian form below:

$$L(p|\mathbf{x}) = H(p|\mathbf{x}) + \lambda * (\sum_k p_k - 1) = -\sum_{k=1}^{K} \psi(p_k) + \lambda * (\sum_k p_k - 1) \tag{23}$$

Taking derivative with respect to $p_k$ and $\lambda$:

$$\frac{\partial}{\partial p_k} L = -\psi'(p_k) + \lambda = 0 \tag{24}$$

$$\frac{\partial}{\partial \lambda} L = \sum_{k=1}^{K} p_k - 1 = 0 \tag{25}$$

Notice that since $\psi(p)$ is *strictly* concave, the function $\psi'(p)$ is monotonically decreasing and therefore invertible. As a result, to solve the maximum entropy problem, we can solve the above systems of equation by finding a inverse function $\psi'^{-1}(p)$, which lead to the simplification:

$$p_k = \psi'^{-1}(\lambda); \qquad \sum_{k=1}^{K} p_k = 1. \tag{26}$$

Above expression essentially states that all $p_k$'s should be equal and sum to 1. The only distribution satisfying the above is the discrete uniform distribution, i.e., $p_k = \frac{1}{K} \ \forall k$. $\qquad\square$

## Footnotes

[4]https://github.com/google/edward2

[5]https://github.com/google-research/bert

[6]https://storage.googleapis.com/bert_models/2020_02_20/uncased_L-12_H-768_A-12.zip