[Reviews · NeurIPS 2020]

Review 1

Summary and Contributions: 1. This paper presents a way to adapt the Gaussian process to high-dimensional data by 1) extracting latent features with a distance-preserving network and 2) introducing distance-aware output layers via random Fourier features, an approximation to the GP. 2. Empirical results on benchmark datasets are provided to show superior performance in out-of-distribution detection.

Strengths: The method presented in the paper can improve out-of-distribution AUC without sacrificing much of the prediction performance.

Weaknesses: 1. The model consists of spectral normalized hidden layers to guarantee a bounded Lipchitz constant for top NN layers, and a random Fourier feature approximated Gaussian process as the last layer. The combination is new, but the overall method is not end-to-end. Thus it can be hard to balance these two components to let them work well with each other. In supplementary material, I saw the entire algorithm is an alternative minimization optimization. So I'm curious about how you choose the initial parameters to make it work well. 2. The main strength is the empirical performance, but the paper does not release code.

Correctness: All techniques are properly used in this paper as far as I can see.

Clarity: 1. I cannot see how Eqn. (5) was used after Section 2. Also, Eqn. (5) itself seems confusing: why your optimal solution p of a minimax problem inf_p sup_{p*} S(p,p*) depends on the inner solution p*? 2. Section 2 seems loosely presented. I suggest presenting the model first and postpone Section 2.

Relation to Prior Work: This paper includes related prior works as far as I can see.

Reproducibility: Yes

Additional Feedback:


Review 2

Summary and Contributions: This paper proposed input-distance-aware metric to measure the uncertainty of deep learning model. The output of DNN is composition of g and h, in which g maps from hidden to the output layer, and h maps from input to the hidden layer. By using Laplacian approximation to Gaussian process which is endowed with distance preserving, the g is distance-aware w.r.t. h. By further using spectural normalization, the h is distance aware w.r.t. x. Combine those together, the distance-aware metric can be derived, to measure the distance of a given sample from training ones, i.e., the extent of uncertainty. This method has been validated on serveral experiments and show promising results on all of them.

Strengths: (i) The authors presented a solid and novel framework to propose a distance-aware metric to measure the uncertainty of a deep learning model. (ii) The idea of spectural normalization which can preserve the order of distance is interesting, not only for uncertainty estimation, but also benefit deep learning model. (iii) This work has conducted convincing experiments on various datasets (including synthetic dataset), which can show the advantage of proposed method, especially the spectural normalization.

Weaknesses: Based on the spectural normalization which bound the l2-norm of W_l, it seems that the \mathcal{X} is with l2-norm as distance metric, which seems not reasonable in image space that may lie in a manifold.

Correctness: Yes

Clarity: Yes

Relation to Prior Work: Yes

Reproducibility: Yes

Additional Feedback:


Review 3

Summary and Contributions: Obtaining accurate uncertainty estimates for deep learning models is a crucial aspect for reliable application fo NNs. This paper proposes a principled method to obtain uncertainty estimates without the need for an expensive sampling process. In essence, the method involves using spectral normalization to regularize the weights of neural networks, and a distance-aware output Layer with the Laplace-approximated neural Gaussian process, in order to promote the input distance awareness, which the authors of the paper identify as a key property for reliable out-of-domain detection. Careful empirical experiments were also conducted to demonstrate the effectiveness of the proposed method under various settings.

Strengths: The paper is very well written. I really enjoyed going through the paper. All in all, in my opinion, this paper is a clear accept to me. The proposed method is sound and reasonable, and the authors of the paper identify a key condition for NN to achieve high-quality uncertainty estimation that was absent in standard DNNs and previously neglected by researchers in the field.

Weaknesses: A major complaint I have is a lack of an ablation study. The proposed method involves two components: the neural GP layer and the spectral normalization. To my understanding, these two components can be implemented independently? What are the relative contributions of the two components? It would be interesting to see such an analysis to really understand the importance of each. Another limitation is the lack of comparison to other baseline algorithms for OOD detection. While the authors of the paper demonstrate the effectiveness of OOD detection compared to other algorithms for uncertainty estimates, there is a lack of comparison against other methods designed explicitly for OOD detection in my opinion.

Correctness: The claims and proposed method seem to be correct in my opinion.

Clarity: The paper is well written overall.

Relation to Prior Work: Prior works are discussed satisfactorily.

Reproducibility: Yes

Additional Feedback:


Review 4

Summary and Contributions: The authors propose a method of uncertainty measure of the deep network prediction. The final layer is replaced by the Gaussian Process (GP) formulation, and the uncertainty information is obtained without multiple forward passes of the same data. The results show that the distance information between data is mainly used for uncertainty measure, while the distance to the prediction boundary is used for other previous methods. It is unclear why the distance to the training data should be used for the uncertainty measure.

Strengths: The method needs one forward pass while the baseline methods in the paper need multiple passes for generating output distribution.

Weaknesses: After reading the paper, it is still arguable why the distance-awareness is decisive for determining the uncertainty. Distance awareness is the property of the conventional local methods such as those using kernels. The experiment for the synthetic data will be reconstructed with conventional Gaussian Processes without neural networks. The effect of the replacement of the last layer seems obvious regarding the distance awareness, but it is unclear whether this distance awareness property is indeed advantageous. From this perspective, the theoretical property in Equation (6) is the property of conventional local methods. It is hard to get the clear idea why the special setting in this paper is inevitable. Deep learning literature has been claiming that the useful transformation of the input space into a disentangled space makes the algorithm powerful, which is contrary to the explanation in this paper that the preserve of the original space distance matters.

Correctness: The experiments for the comparison with conventional methods look fine. But the claim about the advantages of using the proposed method are unclear.

Clarity: The writings and structure of the paper is acceptable.

Relation to Prior Work: The modification from the previous method is clearly explained.

Reproducibility: Yes

Additional Feedback:

[Author Response · NeurIPS 2020]

Thanks to all reviewers for the insightful comments! We have already open-sourced most of our code (link hidden to preserve anonymity) and will provide the link at the beginning of Section 5 of the final paper. Detailed responses below:

**R1: The overall method seems to be not end-to-end.** Algorithm 1 is in fact an end-to-end algorithm. All the trainable parameters $\beta$ and $\{W_l, b_l\}_{l \leq L-1}$ are updated simultaneously in the main SGD step. The only parameters that are not updated are the $W_L$ and $b_L$, which are fixed random features that are frozen throughout the learning. The precision matrix update is only a side computation for summary statistics derived from the SGD-updated model. As a result, the optimization procedure is very similar to the determinisitic SGD training. We use the same initialization for the hidden weights as in the deterministic model, and use the default Glorot uniform initializer for the GP output layer weights $\beta$.

**R1: Clarity regarding Eqn (5).** Thanks for highlighting a point that merits more clarification. Briefly, different from a classic minimax problem, we derived (5) under the assumption that we have partial knowledge about $p^*$ (i.e., we know the domain probability $p^*(x \in X)$), therefore it is possible for the known property of $p^*$ to enter into the final expression. Please see Appendix B for a full statement of the motivation, the assumptions and the proof for Eqn. (5). In particular, please see line 619-626 for an explanation of why (5) is structured as such. Eqn (5) (corresponds to Appendix Proposition 2) is used to motivate Section 2.2. In the final paper, we will include additional explanations / pointers to Appendix B around line 100, and replace "Proposition 2" with "Equation (5)" on line 110 to improve clarity.

**R2: It seems that the $\mathcal{X}$ is with $L_2$-norm as distance metric.** We in fact allow the metric for $\mathcal{X}$ to be non-Euclidean so it reflects the semantically meaningful distance in the data space (please see statement on line 55-56, the discussion on line 141-149, and the proof for Proposition 1 in Appendix D.2. which does not impose restriction on $||.||_\mathcal{X}$ and is based on the theoretical work of [3]). In addition, we'd like to point out that in the vision / language experiments (Section 5.2), the SNGP has superior performance in distinguishing in-domain / out-of-domain data, which is not likely if SNGP can only preserve a $L_2$ metric, which is not suitable for an image / language manifold.

**R3: Ablation study.** We have conducted such ablation study in Appendix C, where DNN-SN and DNN-GP are ablated versions of SNGP. Figure 2-3 shows that in the 2D example, the uncertainty surface of a DNN-SN behaves similarly to a deterministic DNN, while that of a DNN-GP is lacking in preserving input distance. Table 4-6 shows in the vision and language experiments, DNN-SN and DNN-GP tend to outperform the deterministic baseline, but underperform SNGP.

**R3: Comparison to methods designed explicitly for OOD detection.** Please see table for performance comparison to popular OOD methods evaluated using area under precision recall curve (AUPRC) (we will add it to

| Method / AUPRC | C10 vs SVHN | C10 vs C100 | C100 vs SVHN | C100 vs C10 |
|---|---|---|---|---|
| MSP+OE | 89.4 | 76.2 | 52.9 | 32.6 |
| Mahalanobis | 99.1 | - | 98.4 | - |
| ODIN | 92.5 | - | 93.9 | - |
| SNGP | 99.0 | 90.5 | 92.3 | 80.1 |

Appendix C). We denoted CIFAR-10/-100 as C10/100. As shown, despite not designed explicitly for OOD, SNGP is competitive and sometimes outperforms other OOD approaches, especially on difficult near-OOD tasks (e.g, CIFAR 10 v.s. 100 and vice versa). Mahalanobis = Mahalanobis with feature ensemble and inputprocessing.

**R4: Unclear why the distance to the training data should be used for the uncertainty measure / whether this distance awareness property is indeed advantageous.** Intuitively, given a testing example that "looks different" from the training data (i.e., far from the training data manifold), a model's uncertainty metric is expected to return a high value (see, e.g., Fig 1a). Such definition of model uncertainty (or "epistemic" uncertainty) in terms of distance/dissimilarity from observed data has been widely adopted in both the UQ and the ML literature (c.f. Kiureghian and Ditlevsen (2009).Aleatory or epistemic? Does it matter? , Kendall and Gal (2017).What Uncertainties Do We Need in BDL for CV?, *NeurIPS* and the many papers citing them), and empirically can be measured by a model's OOD accuracy (see Table 1-2, and Table 4-6 in Appendix). Quoting other reviewers: "Empirical results on benchmark datasets show superior performance in OOD." (R1), "This work conducted convincing experiments on various datasets...showed the advantage of the proposed method." (R2), etc.

**R4: Why the special setting in this paper is inevitable...Either the theoretical derivations or the empirical results do not support why the practitioners have to use the proposed modification.** Contrary to local methods, vanilla DNN models tend to have difficulty in achieving the distance-preservation property shown in Eqn (6). For example, vanilla DNNs are found to be vulnerable to adversarial examples - they can be sensitive to tiny perturbation in the input space, yet sometimes insensitive to semantics-altering edits to the training data - i.e., not input distance aware [33,34]. To this end, the paper's theoretical result (Proposition 1) ensures SNGP's ability in guaranteeing distance preservation, and the empirical result (Table 1-2, and Appendix C.2) shows that such modification leads to concrete improvement in ECE/OOD performance when compared to an unmodified baseline.

**R4: ...disentanglement...is contrary to the explanation that the distance-preservation matters.** Disentanglement and distance-preservation (i.e. invariance) are both important properties for a representational learning algorithm, and they do not contradict each other (see, e.g., Achille and Soatto (2018).Emergence of Invariance and Disentanglement in Deep Representations, *JMLR*). For a DNN hidden mapping $h : \mathcal{X} \to \mathbb{R}^d$ which is a coordinate transform from the input space to a hidden space $h(x) \in \mathbb{R}^d$, *disentanglement* describes $h(x)$'s ability in separating salient latent features from the noise among its $d$ coordinates, while *distance preservation* describes $h(x)$'s ability in translating a semantically meaningful (often non-Euclidean) measure in the data manifold into that in the Euclidean space [30]. With suitable model specification, disentanglement can happen jointly with distance preservation (e.g., see Figure 2 of [30]). There have been many work that try to achieve both for the purpose of generalization and adversarial robustness, notably via Lipschitz regularization or invertible (i.e. bi-Lipschitz) networks [35, 69] (also, e.g., Engstrom. (2019) Adversarial Robustness as a Prior for Learned Representations)

[Meta-Review · NeurIPS 2020]

3 out of 4 reviewers have accepted as this work as original and offering convincing experiments. However, a knowledgeable reviewer (R4) issued a clear reject. The ensuing discussion over the reason of the reject shows that the meta-reviewer agrees with the concerns of R4, but that the debate this paper triggers may make it worth publishing. This paper offers two clearly distinct algorithms: - one based on Gaussian Processes (GP) builds a loss where the distance between an example and the training data in the last hidden layer is taken into account for OOD modelling - one based on Spectral Norm (SN) better ties the distance in the hidden space to the input space distance. This is justified by Lipschitz bounds that seem very loose. The objections raised by R4, but also hinted by other reviewers are serious: in a deep learning architecture, as the input data lives in a low dimensional manifold, there is no reason for a distance that is not aware of this manifold to be meaningful (except locally as shown for adversarial learning). Many distance-based methods for OOD look at the activations in penultimate layer and do not justify this from a mapping to the input layer. However, experiments reported in the appendix (section C.2, table 6) show that the SN algorithm is essential for performance. While I agree with R4 that it is unlikely this algorithm can be properly justified by the loose Lipschitz bounds, like Batch Normalization, the authors may have stumbled into a very powerful algorithm with an unsatisfactory explanation. Acting in the opposite direction as Batch Normalization, this algorithm seems to reduce the range of weights and activations and improve calibration.